



**Ultradian rhythms in shell composition of photosymbiotic and non-photosymbiotic**
**mollusks**
Niels J. de Winter[1,2], Daniel Killam[3], Lukas Fröhlich[4], Lennart de Nooijer[5], Wim Boer[5], Bernd R.
Schöne[4], Julien Thébault[6], Gert-Jan Reichart[2,5]
*Affiliations*
[1]Analytical, Environmental and Geochemistry group (AMGC), Vrije Universiteit Brussel, Brussels,
Belgium
[2]Dept. of Earth Sciences, Utrecht University, Utrecht, the Netherlands
[3] San Francisco Estuary Institute, Richmond, CA, USA
[4] Department of Paleontology, Institute of Geosciences, Johannes Gutenberg Universität, Mainz,
Germany
[5] Dept. of Ocean Systems, Royal Netherlands Institute for Sea Research (NIOZ), Texel, the
Netherlands
[6]Univ Brest, CNRS, IRD, Ifremer, LEMAR, 29280 Plouzané, France, (ORCID: 0000-0002-3111-

16 4428)

Corresponding author: N.J. de Winter, niels.de.winter@vub.be



**Abstract**
The chemical composition of mollusk shells is a useful tool in (paleo)climatology since it captures
inter- and intra-annual variability in environmental conditions. Trace element and stable isotope
analyses with improved sampling resolution now enable the use of mollusk shells for
paleoenvironmental reconstructions at a daily to sub-daily resolution. Here, we discuss hourly
resolved Mg/Ca, Mn/Ca, Sr/Ca and Ba/Ca profiles measured by laser ablation ICP-MS through
shells of photosymbiotic giant clams (*Tridacna maxima*, *Tridacna squamosa* and *Tridacna*
*squamosina*) and the non-photosymbiotic scallop *Pecten maximus*. Precise sclerochronological
age models and spectral analysis allowed us to extract daily and tidal rhythms in the trace element
composition of these shells. We find significant expression of these periodicities but conclude that
this cyclicity explains less than 10% of the sub-annual variance in trace element profiles. Tidal
and diurnal rhythms explain variability of at most 0.2 mmol/mol (~10% of mean value) in Mg/Ca
and Sr/Ca, while Mn/Ca and Ba/Ca cyclicity has a median amplitude of less than 2 μmol/mol
(~40% and 80% of the mean of Mn/Ca and Ba/Ca, respectively). Daily periodicity in Sr/Ca and
Ba/Ca is stronger in *Tridacna* than in *Pecten*, with *Pecten* showing stronger tidal periodicity. One
*T. squamosa* specimen which grew under a sunshade exhibits some of the strongest diurnal
cyclicity. Daily cycles in trace element composition of giant clams are therefore unlikely to be
driven by variations in direct insolation itself but reflect an inherent biological rhythmic process
affecting element incorporation. Finally, the large amount of trace element variability unexplained
by periodic variability highlights the dominance of aperiodic processes in mollusk physiology
and/or environmental conditions on shell composition at the sub-daily scale. Future studies should
aim to investigate whether part of this aperiodic variability in shell chemistry reliably records
weather patterns or circulation changes in the paleoenvironment.
**1. Introduction**





Patterns in growth increments, microstructure, and chemical composition of accretionary
carbonate bioarchives yield detailed information about the environmental conditions and
biological rhythm of carbonate producing animals (Dunbar and Wellington, 1981; Jones, 1983;
Witbaard et al., 1994; Klein et al., 1996; Surge et al., 2001; Schöne et al., 2005a; Ivany, 2012;
Schöne and Gillikin, 2013; DeCarlo and Cohen, 2017; Killam and Clapham, 2018). These
characteristics have spurred the development of a multitude of techniques for extracting
information about life history (Jones and Quitmyer, 1996; Schöne et al., 2005b; Goodwin et al.,
2009; Mahé et al., 2010; Comboul et al., 2014; DeCarlo and Cohen, 2017; Judd et al., 2018; de
Winter, 2022), carbonate chemistry (Sinclair et al., 1996; Lazareth et al., 2003; Schöne et al.,
2010; de Winter and Claeys, 2017; Warter and Müller, 2017; Huyghe et al., 2021; de Winter et
al., 2021a) and microstructure (Lazier et al., 1999; Checa et al., 2007; Popov, 2014; Gilbert et al.,
2017; Crippa et al., 2020; Höche et al., 2020; 2021; Wichern et al., 2022) from carbonate shells
and skeletons. As a result, (fossil) carbonate skeletons have gained much attention as archives
of past environmental and climate change (e.g., Lough, 2010; Schöne and Gillikin, 2013; Ivany
and Judd, 2022 and references therein).
Three characteristics make the shells of marine mollusks especially valuable as climate archives:
(1) Nearly all marine mollusks precipitate their shells in isotopic equilibrium with ambient sea
water, except for juvenile oysters and some mollusks growing near hydrothermal vents (Schöne
et al., 2004; Wisshak et al., 2009; Huyghe et al., 2021; de Winter et al., 2022), (2) mollusk shells
have a high fossilization potential and long geological history, dating back to the beginning of the
Phanerozoic (Al-Aasm and Veizer, 1986a; b; Jablonski et al., 2003; Cochran et al., 2010;
Jablonski et al., 2017; de Winter et al., 2017; 2018; Coimbra et al., 2020), (3) the incremental
growth of mollusk shells allows for internal dating within the shell, yielding chronologies of shell
growth with sub-annual precision (Richardson et al., 1980; Jones, 1983; Schöne et al., 2005c;
Goodwin et al., 2009; Huyghe et al., 2019). These advantages enable mollusk shells to record



important information about climate and ambient water chemistry on the seasonal scale. Thereby,
reconstructions from mollusk shells are highly complementary to other, less highly resolved but
longer-term, climate and environmental reconstructions like sedimentary records, tree rings and
ice cores (Black, 2009; Bougeois et al., 2014; Petersen et al., 2016; Tierney et al., 2020; de Winter
et al., 2021b).
The resolution of the mollusk shell archive is not limited to seasonal variability. Studies monitoring
the behavior of mollusks during growth experiments show that their activity varies as a function
of environmental conditions (e.g., temperature and food availability) and follows ultradian rhythms
which may contain daily to hourly periodicities, probably linked to diurnal and tidal cycles, or lack
periodic behavior altogether (Rodland et al., 2006; García-March et al., 2008; Tran et al., 2011;
Ballesta-Artero et al., 2017; Xing et al., 2019; Tran et al., 2020). Analyses of growth patterns and,
more recently, composition of shell carbonate deposited at these short time intervals show that
these rhythms can be recorded in mollusk shells (Pannella, 1976; Richardson et al., 1980; Sano
et al., 2012; Warter et al., 2018; de Winter et al., 2020). This raises the question whether mollusk
shells reliably record behavioral changes, high frequency (paleo-) weather or circulation patterns
(e.g., Komagoe et al., 2018; Yan et al., 2020; Poitevin et al., 2020). Alternatively, the presence of
daily cyclicity in shell chemistry may yield information about the paleobiology of extinct mollusks,
such as the presence of photosymbiosis (e.g., Sano et al., 2012; Warter et al., 2018; de Winter et
al., 2020). The latter seems plausible given the effect of photosymbiosis on shell mineralization
in modern tridacnids (Ip and Chew, 2021) and on the trace element composition of aragonite in
modern photosymbiotic scleractinian corals (Cohen et al., 2002; Meibom et al., 2003; Inoue et al.,
2018). If proven true, daily variability in bivalve shells may serve as a proxy for photosymbiosis in
the fossil record (e.g., de Winter et al., 2020). This is of interest because photosymbiosis is a
derived adaptation of some tropical bivalve species (e.g., tridacnids) and its prevalence in the
fossil record has important implications for the ecological niche of fossil mollusks (e.g., Vermeij,





2013). In addition, photosymbiosis can affect mollusk shell composition, and understanding it is
therefore critical for the interpretation of chemical proxies in mollusk shells for environmental
reconstructions (Killam et al., 2020). Finally, improving our understanding of photosymbiosis in
tropical ecosystems sheds light on the resilience of photosymbiotic organisms to environmental
change, now and in the geological past. The latter is of special interest in light of the ongoing
climate and biodiversity crises, which are profoundly affecting these sensitive ecosystems
(Pandolfi and Kiessling, 2014).
In this study, we investigate shell growth patterns and shell chemistry of the photosymbiotic
bivalves *Tridacna maxima*, *T. squamosa* and *T. squamosina* as well as the non-photosymbiotic
scallop *Pecten maximus*. *P. maximus* was chosen as a non-photosymbiotic counterpart in
comparison with the tridacnids because of its comparatively high growth rate and the presence of
daily striae on the outside of its shell, which make it possible to construct accurate shell
chronologies (Chauvaud et al., 2005). We combine ultra-high-resolution (hourly resolved) Mg/Ca,
Sr/Ca, Mn/Ca and Ba/Ca measurements in the shells with detailed sclerochronology to investigate
the variability in these trace elements over time in all four species. The aim of this study is to
investigate (1) whether the shells record high-frequency variability in shell chemistry that can be
linked to environmental and/or circadian rhythms and (2) whether the presence of photosymbiosis
influences the expression of this variability in the shells' composition.



## 2. Materials and methods

2.1 Preparation of *P. maximus* specimens

Three specimens of the King scallop *P. maximus* (labeled "**PM2**", "**PM3**" and "**PM4**") were collected alive on 15/11/2019 on the southern coast of the Bay of Brest near Lanvéoc, France (48°17′N 4°30′W) by SCUBA divers at a depth of approximately 8 m (see Fröhlich et al., 2022; **Figure 1**). Note that water depth in the Bay of Brest varies significantly due to the macrotidal regime with a mean tidal range of 2.8 – 5.9m with extreme ranges up to 7.2m (Guillaume-Olivier et al., 2021; Service Hydrographique et Océanographique de la Marine; 2022). Collected specimens contained at least one full year of growth based on the visibility of one winter growth line on the outside of the shell (age class 1; see Thébault et al., 2022 ; **Fig 1F** and **S1**). Specimens were frozen at -20°C immediately after collection. Soft body parts and epibionts were removed from the shells before further treatment. Shells were superficially cleaned using a plastic brush and adhering sediment was removed by ultrasonication in deionized water. The flat, left valves were used for elemental and sclerochronological analysis following previous studies on *P. maximus* (Thébault et al., 2022; Fröhlich et al., 2022).

High-resolution color photos were made of the outside of the left valve of the shell using a mirror-reflex camera (Canon EOS 600 DSLR camera connected to a Wild Heerbrugg binocular microscope equipped with a Schott VisiLED MC 1000 light source) aimed downward perpendicular to the working surface. Overlapping images of the shells were stitched together using Image Composite Editor v2.0.3.0 (Microsoft Research Computational Photography Group, Redmond, WA, USA). The stitched images were used to count and measure daily striae on the shell surface (see **Fig. 1** and **S1**). To obtain a fully focused composite of the complete shell, dynamic focusing was applied to allow all parts of the slightly curved surface of the shell to come into focus. Dynamic focus images where later stitched together using focus stacking in Helicon Focus (Helicon Focus 7.7.5; HeliconSoft, Kharkiv, Ukraine; see **S1**).



Cross sections were cut through all three *P. maximus* shells perpendicular to the daily growth
lines (striae) from the ventral margin of the shell to the shell hinge (see **Fig. 1B**, **Fig. 1E-F** and
**S1**) along the axis of maximal growth. Shells were fortified with a protective layer of metal epoxy
(Gluetec Wiko Epofix 05) before sectioning using a Buehler Isomet 1000 low-speed precision saw
(Buehler Inc, Lake Bluff, IL, USA) equipped with a diamond-coated wafering thin blade (0.4 mm
thickness; number 15LC 11–4255) at 200 rpm. Parallel cuts were made to allow shell sections to
be glued to glass plates for high-grade polishing (down to F1200 grit SiC powder and 1 µm $Al_2O_3$
suspension). Two cross sections were made through specimens **PM2** and **PM3**: One through a
"rib" of the shell (i.e., radial segment that protrudes away from the interior, named **PM2_1** and
**PM3_1**) and one through a "valley" (i.e., radial segment between two "ribs" that lies deeper
towards the interior, named **PM2_2** and **PM3_2**; see **Fig. 1** and **S1**). The dual sections were cut
to compare shell chemistry between the "ribs" and "valleys" of the shell . Specimen **PM4** was only
sectioned once, through a "valley" in the shell, making a total of five cross sections through the *P.*
*maximus* specimens.





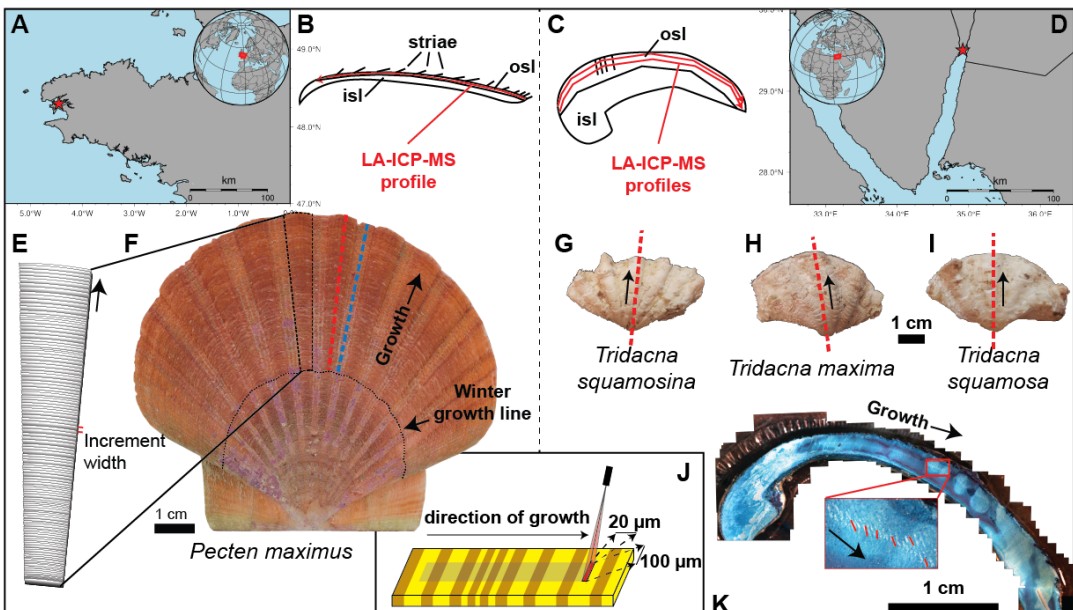


**Figure 1**: **Overview of sample locations and preparation steps**. **A)** Location of the Bay of Brest, with the red star indicating the sampling location. **B)** Schematic cross section through *P. maximus* showing how the LA-ICP-MS linescan (red line) was positioned within the outer shell layer (OSL). **C)** Schematic cross section through a tridacnid, illustrating the positions of parrallel LA-ICP-MS line scans (red lines) through these shells within the OSL. **D)** Position of the Gulf of Aqaba, with the red star indicating the sample location for tridacnids. **E)** Schematic representation of a segment through the shell of *P. maximus* showing the striae which are deposited daily and which were counted to establish age models (see also **B**). **F)** Left valve of *P. maximus* (**PM2**) used in this study, with dashed lines showing the position of cross sections through ribs (red) and valleys (blue) in the shell. Black arrow indicates growth direction away from the shell hinge. The black dotted line highlights a winter growth stop. **G-I)** Pictures of (from left to right) *T. squamosa* (specimen **TSFRS1**), *T. maxima* (specimen **TM29**) and *T. squamosina* (specimen **SQSA1**) with dashed red lines indicating the positions of the cross sections used for LA-ICP-MS analysis (see **C**) and black arrows indicating the direction of growth. **J)** Schematic representation of the LA-ICP-





MS line scanning setup with the rectangular spot size (100 * 20 μm; see **S11**) that was positioned
parallel to the growth layers in the shell. **K**) Example of Mutvei-stained cross section through a *T.*
*maxima* specimen used to visualize and count growth lines, with the insert showing part of the
OSL where growth lines were counted (red lines) to establish age models for the tridacnids. Black
arrows indicate the direction of growth.





2.2 Preparation of *Tridacna* specimens
A total of 5 tridacnid specimens, two *T. maxima* (named **TM29** and **TM84**), two *T. squamosa*
(named **TS85** and **TSFRS1**) and one *T. squamosina* (**SQSA1**) specimen, were collected in the
summer of 2016 from beach death assemblages on the coast of the Gulf of Aqaba with permit
from the Israeli National Parks Authority (**Figure 1**; see details in Killam et al., 2020). One cultured
*Tridacna squamosa* shell (**TSM1**) was obtained from the National Center for Mariculture, Eilat.
Species were determined following shell characteristics of the local population as cited in Roa-
Quiaoit (2005).
All shells were sectioned along the axis of maximum growth after removing epibionts using a
metal brush (see **Fig. 1G-I**). Original microstructure and preservation of the original aragonite
mineralogy of all specimens was confirmed using Scanning Electron Microscopy and X-ray
Diffraction Spectroscopy following Gannon et al. (2017) and Kontoyannis and Vagenas (2000;
see details in Killam et al., 2020). Shell segments were partially embedded in Araldite 2020 epoxy
resin (Huntsman Corp., Woodlands, TX, USA) before being sectioned in direction of maximum
growth using a slow-rotating saw equipped with a thin wafered saw blade (ø < 1 mm). Parallel
cross sections produced 5-10 mm thick sections that were polished using progressively finer SiC
polishing disks.

2.3 Microscopy and photography
Polished surfaces of all 11 cross sections (5 *Pecten*, 6 *Tridacna*) were imaged using an Epson®
1850 flatbed scanner (Seiko Epson Corp., Nagano, Japan) at a pixel resolution of 6400 dpi (±4
µm pixel size) as well as by stitching micrographs made using a KEYENCE VHX-5000 digital
microscope using x250 magnification together into composite images (see **S1**). Cross sections





were imaged both before and after trace element analyses to allow the trace element profiles to
be referenced relative to the cross sections.

2.4 LA-ICP-MS analyses
Elemental ratios were based on measuring ratios of the isotopes $^{25}$Mg, $^{87}$Sr, $^{55}$Mn and $^{137}$Ba to
$^{43}$Ca along profiles through all shell cross sections using Laser Ablation – Inductively Coupled
Plasma – Mass Spectrometry (LA-ICP-MS). Measurements were carried out on a laser ablation
system (ESI NWR193UC; Elemental Scientific, Omaha, NE, USA) coupled to a quadrupole ICP-
MS (iCap-Q, Thermo Fisher Scientific, Waltham, MA, USA) at the Royal Netherlands Institute for
Sea Research (NIOZ). Operation parameters are provided in **S11**.




Scan lines were programmed on the polished shell cross sections in direction of growth as close
as possible to the outer edge of the shell, with the LA-ICP-MS spot oriented parallel to the growth
lines (with a width of 20 µm in scanning direction, see **Fig. 1J**; **S11**). For the pectinids, care was
taken to target the outer portion of the outermost shell layer (oOSL) and avoid sampling of the
inner portion of the outer shell layer (iOSL) or inner shell layer (ISL), which was demonstrated to
have a different chemical composition (see Freitas et al., 2009). For the tridacnids, profiles were
placed within the OSL close to (within 100 µm of) the outer edge of the shell in a first analytical
session. However, since spikes of high Mg/Ca and Mn/Ca ratios were observed in these results,
parallel transects placed ~100 µm further towards the inside of the shell were measured through
all tridacnid shells to verify whether these spikes in Mg and Mn were reproducible further inward
(see **S2**). All scan lines in pectinids and tridacnids were repeated a second time at the exact same
location using a faster scan rate of 10 µm.s$^{-1}$ to assess repeatability of the elemental signals (see
**S2**).
Data reduction was performed using an adapted version of the data reduction software SILLS
(Signal Integration for Laboratory Laser Systems; Guillong et al., 2008) in Matlab. Raw LA-ICP-
MS data were calibrated using NIST610, (National Institute of Standards and Technologies,
Gaithersburg, MD, USA) using the reference values reported in the GeoReM database (Jochum
et al. 2005, 2011). Quality control materials BAS752 (Bureau of Analyzed Samples,
Middlesbrough, UK), RS3 and one matrix-matched carbonate standard (MACS-3; United States
Geological Survey, Reston, VA, USA; Wilson et al., 2008) were used to monitor the quality of the
measurement. To increase the stability of the ICP-MS signal and to correct for drift, $^{43}$Ca was
used as internal standard. External drift-correction using repeated measurements on the JCp1
standard was applied if the element/Ca drift was >5% during the analytical sequence. Drift during
a single transect was found to be negligible.




## 2.5 Age models

Trace element profiles in *P. maximus* shells were internally dated using daily striae visible on the outer shell surface (**Fig. 1E**). Daily increment widths (perpendicular distances between successive striae) were counted and measured multiple times, both on the outside of the shell using the focus-stacked images (see **section 2.3**) and by counting and measuring the distance between growth layers in cross sections through the "valleys" of the shells (**PM2_2** and **PM3_2**; see **S3**) by different persons. Positions of daily striae on the outside of the shells were plotted relative to distance along the LA-ICP-MS scan line using manual alignment of striae and the LA-ICP-MS path on microscope composites of cross sections through the shells, taking into account the curvature of growth lines with distance away from the outer shell surface (see **S3**). The timing of shell formation was determined by backdating the daily striae from the ventral margin (last visible stria mineralized on the date of shell collection, i.e., November 15, 2019), and by linearly interpolating the timing of measurements located between daily growth lines based on their distance from daily striae positions (**S5**).

Trace element profiles from *Tridacna* shells were also dated using layer counting. However, since expression of daily and semi-diurnal growth markings was insufficiently clear to count individual growth lines along the full (multi-year) growth period recorded in all the shells, age models were constructed based on parts of the shell where daily and tidal layers could be identified with confidence. Polished cross sections through all tridacnids were imaged using UV luminescence (see **Fig. 1K** and **S4**) to facilitate this counting. The median widths of daily or semi-diurnal increments were determined on these cross sections and compared to the width of annual increments identified based on growth breaks visible on the outer margin of the shell. The distinction between diurnal (24h) and tidal (~12h) pacing of growth increments was made based on the width of small-scale increments relative to the width of annual increments in the shell. A



von Bertalanffy growth model (Von Bertalanffy, 1957) was constructed for each specimen based
on the annual growth (ΔL) inferred from growth line counting and the maximum shell height ($L_{inf}$)
known for these species in the Red Sea from the literature (Roa-Quiaoit, 2005; Mohammad et al.,

262   2019):

$$L_t = L_{inf} * \left(1 - e^{-kt}\right), with k = -ln\left(\frac{\Delta L}{L_{inf}}\right)$$
In this formula, $L_t$ is the shell height at time t and k is the growth constant (Brody growth coefficient;
Munro, 1984). Since cross sections through the tridacnids were made through the shell hinge (in
direction of the shell height) and literature values for $L_{inf}$ are reported with reference to shell length
(measured parallel to the shell hinge), allometric data on *T. maxima*, *T. squamosa* and *T.*
*squamosina* from the literature was used to convert $L_{inf}$ values (which are commonly reported as
shell length) to shell height and make them relevant for the direction in which the trace element
profiles were measured on the cross sections (Roa-Quiaoit, 2005; Richter et al., 2008;
Mohammad et al., 2019). Uncertainties on the annual growth increment widths (ΔL) were
calculated from the standard error of the mean width of daily and semi-diurnal growth increments
on which ΔL is based, and uncertainties on the values for $L_{inf}$ were taken from variability in the
values in the literature. Both sources of uncertainty were propagated through the growth model
using the variance formula (Ku, 1966) to obtain error envelopes on age-distance relationships
(growth curves) of tridacnids (see **S5**). All data processing steps described in this manuscript are
carried out using the open-source computational software package R (R Core Team, 2022), and
scripts detailing these calculations are provided in **S6** and deposited on the open-access software
repository GitHub (https://zenodo.org/record/6603175)

2.6 Spectral analysis





Spectral analysis on the LA-ICP-MS data was used to isolate trace element variability at the sub-
annual scale. All trace element profiles were first detrended using a LOESS filter with a span of
0.2 times the length of the record to remove longer term (i.e., seasonal to multi-annual) trends.
The detrended series were linearly resampled in the time domain before applying the Multi-Taper
Method (MTM; Thomson, 1982) to extract dominant frequencies from the data. Spectral analysis
was carried out using the "astrochron" package (Meyers, 2014) in R (R Core Team, 2022; see
script in **S6**). The significance of relevant periodicities was tested using a combination of "red
noise" estimation and a harmonic F-test (see Meyers, 2021). To visualize the evolution of periodic
behavior across the shells, wavelet analysis was applied on all trace element profiles using the
"dplR" package in R (see **S6**).

2.7 Extracting high-resolution variability
After detrending and spectral analysis, all trace element profiles were smoothed using a Savitzky-
Golay filter with a width of 21 datapoints (8.4 µm; equivalent to a timespan of ~1-5h; **S6**) to remove
high-frequency measurement noise. Statistically significant (see **section 2.6**) variability in daily
(~22-36h; centered on the 24h diurnal cycle) and tidal (~8-14h; centered on the 12.4h tidal cycle)
frequency bands was extracted from the trace element records using a combination of bandpass
filtering (using the "bandpass" function in the "astrochron" R package) and stacking (see **S6**).
Trace element data was stacked along bandpass filters using the following procedure: Maxima
and minima in the bandpass filter were used as tie points to reference each datapoint of the
smoothed dataset relative to its position within the cycle on a scale from 0 to 1. These relative
positions were then used to divide the data into 10 bins (bin 1 contains positions 0 – 0.1, bin 2
contains data from positions 0.1 – 0.2, etc.), giving the stacked data a resolution of 0.1 times the
length of the cycle under investigation. The full breakdown of variability within and between bins
created in the stacking routine is provided in **S7**. Different sources of variance in the trace element



records were isolated by sequentially determining the variance left in the trace element records
after each of the data treatment steps explained above (see example in **S7**). This procedure
allowed us to quantify the amount of variance in each trace element profile explained by either
diurnal or semi-diurnal variability.





**3. Results**
3.1 Trace element data
LA-ICP-MS line scans yielded profiles of Sr/Ca, Mg/Ca, Mn/Ca and Ba/Ca in growth direction on
11 cross sections through shells of *P. maximus*, *T. maxima*, *T. squamosa* and *T. squamosina*.
Trace element profiles of consecutive line scans on the same transect show high repeatability:
sub-millimeter scale patterns in Sr/Ca, Mg/Ca, Mn/Ca, and Ba/Ca are repeated between
consecutive line scans, $R^2$ values between trace element results of time-equivalent shell samples
typically exceed 0.8, and the mean ratio difference between time equivalent samples in different
line scans is less than 0.05 mmol/mol for the most variable profiles (Mg/Ca, with lower differences
for the lower-concentration Mn/Ca and Ba/Ca records; see **S2**). Remeasured transects further
away from the outer shell surface in tridacnids (see **section 2.4**) differ more from the original
transects than those measured on the exact same locality in the shell: $R^2$ values between parallel
lines in different localities are 0.3 – 0.5 for Mg/Ca and Sr/Ca and <0.3 for Mn/Ca and Ba/Ca,
reflecting intra-shell variability in trace element composition in the tridacnids (**S2**). Overall, sub-
millimeter scale patterns in trace element composition are reproduced in parallel line scans, and
the mean offset between the lines was always less than 0.2 mmol/mol.
Pectinid and tridacnid shells contain similar mean Sr/Ca and Ba/Ca ratios (Sr/Ca of 1.3 ± 0.3 and
1.5 ± 0.6 mmol/mol respectively; Ba/Ca of 2.8 ± 2.5 and 3.0 ± 5.1 μmol/mol respectively;
uncertainty is calculated as 1σ). Mean Mg/Ca and Mn/Ca ratios are higher in *P. maximus* than in
*Tridacna* species (Mg/Ca = 3.1 ± 0.9 and 0.7 ± 0.9 mmol/mol; Mn/Ca = 7.8 ± 4.7 and 2.7 ± 7.8
μmol/mol; 1σ; **Figure 2**; **S4**). Differences between tridacnid specimens generally exceed the
differences between tridacnids and pectinids (1σ of Ba/Ca among all tridacnid specimens = 2.1
μmol/mol). Individual records like those in **TM84** and **PM3_1** show large variability (especially in
Ba/Ca and Mn/Ca) compared to other specimens of the same species. Inter-specimen variability
is higher in tridacnid shells than in pectinids (inter-specimen relative standard deviations as a




fraction of mean ratio for Ba/Ca: 0.74 vs 0.64, Mg/Ca: 0.37 vs 0.20, Sr/Ca: 0.19 vs 0.03 and
Mn/Ca: 0.78 vs 0.33 for tridacnids and pectinids, respectively). **Figure 2** shows that this variability
between tridacnids is not readily explained by differences between species, but mostly reflects
differences in the trends within the records, with some specimens (e.g., **TM84**, **TSM1** and **TS85**)
showing trends in composition towards the end of the record (see also **S8**). Trace element
compositions in tridacnid shells are significantly more skewed towards higher values than in
pectinids (mean skewness per element and per specimen is 9.7 for tridacnids and 0.9 for
pectinids), reflecting the high peaks in trace element composition observed in tridacnid profiles,
especially near the ventral margin (e.g., specimens **TM84**, **TSM1** and **TS85**; see **section 2.4**; **Fig.**
**2**; **S8**). Finally, "rib" and "valley" segments through the same specimen of *P. maximus* show similar
patterns in trace elements, but absolute concentrations (especially of Ba and Mn) can be quite
different, highlighting heterogeneity within the shells of *P. maximus* (**Fig. 2**).
Plots of trace element variability reveal dominant high-frequency variability superimposed on
seasonal-scale patterns (**Figure 2**). Trace element profiles in pectinids, reflecting only one
growing season, show a typical seasonal pattern in Sr/Ca and Mg/Ca with maxima in the
elemental ratio in the middle of the profile (corresponding to the summer). Mn/Ca and Ba/Ca in
pectinids are more variable, showing multiple peaks in the same growth year. Peaks in Mn/Ca
and Ba/Ca are synchronous between profiles through the same specimen, but not between
specimens, possibly showing that growth resumed on different days for different specimens after
the winter stop. Like in the pectinid profiles, Mg/Ca, and Sr/Ca ratios in tridacnids show similar
patterns, with one or two distinct cycles per growth year. However, higher frequency variability in
tridacnid ratio profiles is characterized by more extreme peaks, especially in Mg/Ca, skewing the
distribution of trace element values. Mn/Ca and Ba/Ca appear to be less variable in tridacnid
shells than in pectinids, except for specimen **TM84**, which shows a sharp increase in Mn and Ba
towards the end of its lifetime. Mn/Ca and Ba/Ca ratios in tridacnids show more regular annual or



biannual variability than pectinids (most notably specimen **SQSA1**). It must be noted, however,
that *P. maximus* shells only recorded one growth season, limiting the interpretation of seasonal
growth patterns.



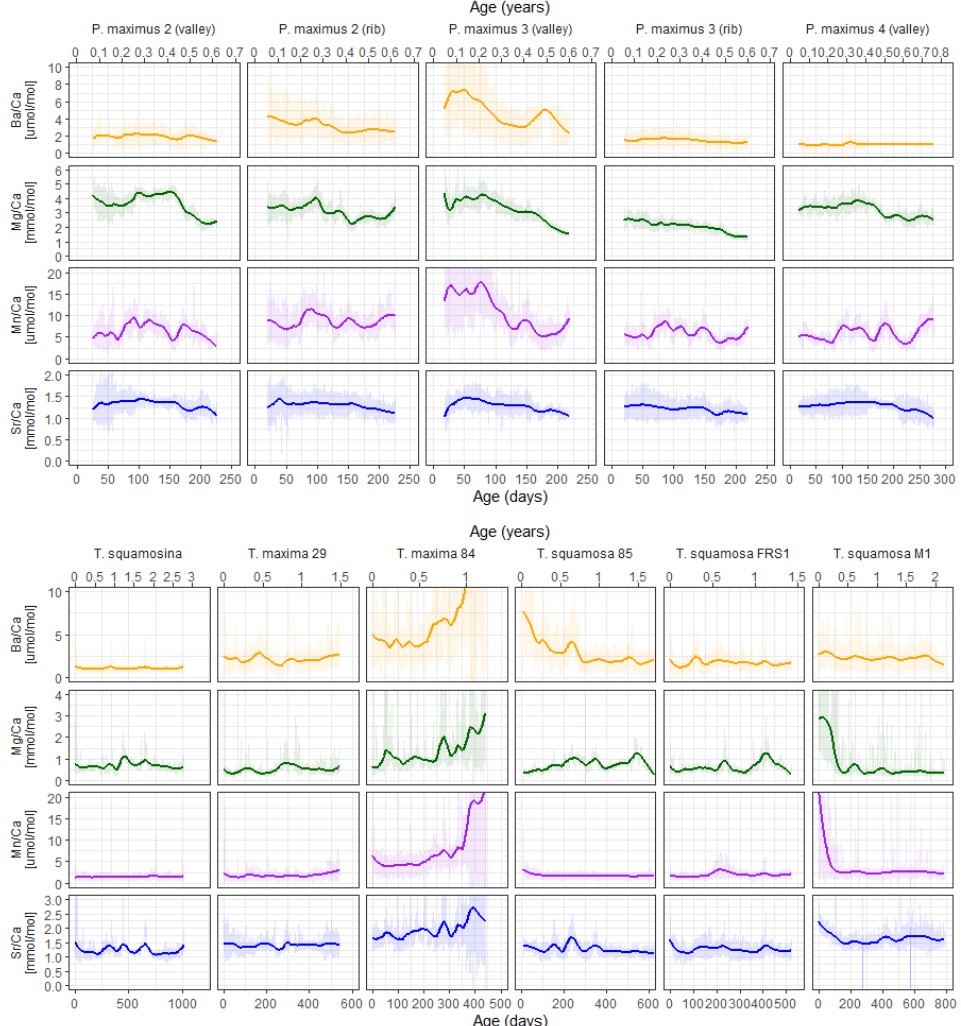


**Figure 2**: Overview of LA-ICP-MS results of Sr/Ca (blue), Mg/Ca (green), Mn/Ca (purple) and

Ba/Ca (orange) in pectinid (upper panel) and tridacnid (lower panel**)** specimens. Vertical axes are

equal for plots positioned next to each other (but different for the two groups of tridacnid and

pectinid plots). Shaded lines show raw LA-ICP-MS data while solid lines indicate 0.2 span LOESS

fits through the data highlighting monthly-scale variability. A direct comparison of trace elemental

ratios between specimens is provided in **S8**.





3.2 Age models
Growth line counting in the *P. maximus* shells was repeated multiple times on both the outer shell
surface and in cross sections through the shell by different persons (**Table 1**; **S3**). The variability
in counting results shows that the growth lines were not always equally easy to distinguish. In
**PM2** and **PM3**, the most likely number of increments (228 and 220 respectively) was counted in
both cross sections and on the outside of the shell, with other analyses yielding both higher and
lower numbers. In **PM4**, counts on the outside and on the one available cross section were very
close (**Table 1**). In this case, the counting in the cross section (278 increments) was chosen as
reference since the LA-ICP-MS profile was measured on the same cross section and could be
directly linked to the counted increments. The fact that the mean increment width between the *P.*
*maximus* specimens which grew in the same year in the same environment is highly consistent
lends confidence to the layer counting result (**Table 1**). The difference in number of days of growth
between specimens can be caused either by variability in the day on which seasonal growth
commenced (in spring) or the day on which the winter growth stop commenced (in autumn;
Chauvaud et al., 1998). The sampling date (November 15[th], 2019) does not preclude the onset of
winter growth cessation before the time of sampling. The age-distance relationships (growth
curves) resulting from the sclerochronology are shown in **S5**.



**Table 1: Growth increments counting in *P. maximus***

| Specimen | Increments counted on outer surface | Increments counted in cross sections | Mean increment width [μm ± 1σ] |
|---|---|---|---|
| **PM2** | 226, **228**, 234, 241 | 227, **228**, 233 | 249 ± 19 |
| **PM3** | **220**, 226, 243 | 213, **220**, **220** | 249 ± 22 |
| **PM4** | 272, 273 | **278** | 247 ± 4 |







Layer counting in tridacnid shells yielded estimates of semi-diurnal, daily and annual growth
(**Table 2**; **S4**). Annual growth rates calculated from layer counting are highly consistent between
specimens from the same species from the same environment, lending confidence to the growth
line counting results. The von Bertalanffy growth models based on these growth line countings
are plotted in **S5**. Statistics of the parameters ($L_{inf}$ and k) of these growth models and their
uncertainty are provided in **S4**.



**Table 2: Growth line counting in *Tridacna* shells.** Column 3 shows the total number of
increments counted in the specimen, column 4 shows their median width and column 5 shows
the width of an annual increment in the specimen. Note that increments could not be counted over
the entire growth period of the shells, so the numbers in column 3 represent representative
numbers of increments counted in those parts of the shells where they were distinct enough for
counting (see **section 2.5**) Increment timing (semi-diurnal vs diurnal) was established based on
the relative difference between small increment width and annual increment width.

| Specimen | Species | # counted increments | Median increment width [µm] | Annual growth [mm] | Increment timing |
|---|---|---|---|---|---|
| **TM29** | *T. maxima* | 274 | 26.5 | 27.9 | Semi-diurnal |
| **TM84** | *T. maxima* | 109 | 39.1 | 26.6 | Diurnal |
| **TS85** | *T. squamosa* | 310 | 40.3 | 20.2 | Diurnal |
| **TSFRS1** | *T. squamosa* | 225 | 23.3 | 20.1 | Semi-diurnal |
| **TSM1** | *T. squamosa* | 180 | 33.3 | 20.6 | Diurnal |
| **SQSA1** | *T. squamosina* | 153 | 22.3 | 14.9 | Diurnal |





Growth rates are highly similar between specimens of the same species (**Table 1 and Table 2**;
**S3-5**), with *P. maximus* achieving the highest growth rates (~220 growth days * ~250 µm/d ≈ 55
mm/yr; **Table 1**), followed by *T. maxima* (~27 mm/yr; **Table 2**), *T. squamosa* (~20 mm/yr; **Table**
**2**) and *T. squamosina* (15 mm/yr; **Table 2**). The age models reveal that the average temporal
resolution of the LA-ICP-MS line scans was 0.04h, 0.24h, 0.44h and 0.27h for *P. maximus*, *T.*
*maxima*, *T. squamosa* and *T. squamosina*, respectively. These estimates were calculated by
dividing the width of the daily increments (e.g., 250 µm for *P. maximus*) by the resolution of the
LAICPMS data (0.4 µm) to obtain the number of LAICPMS measurements per day (e.g., 625
pts/day for *P. maximus*, yielding a mean sampling resolution of 0.04h). Note that the LA-ICP-MS
slit is wider (20 µm) than the spatial sample resolution, causing some smoothing on the scale of
this very fine temporal resolution. The LA-ICP-MS profiles record trace element variability during
growth periods ranging between 220 days (for **PM3**) and 1041 days (for **SQSA1**).

3.3 Spectral analysis
Normalized power spectra and significance level of daily and tidal periodicities in pectinid and
tridacnid records are shown in **Figure 3** and **Figure 4**, respectively. Full spectral analysis results
for all trace element records in all specimens are provided in **S9**. All *P. maximus* power spectra
(**Fig. 3**) reveal semi-diurnal (12h) periodicity in Sr/Ca and Ba/Ca with >86% statistical significance.
Only sections through the ribs of the shells (**PM2_1** and **PM3_1**) show semi-diurnal periodicity in
Mg/Ca and Mn/Ca (>90% significance). Daily periodicity is present in some pectinid profiles, but
there seems to be no consistent pattern in the presence of diurnal variability between specimens,
between sections through ribs or valleys in the shell or between trace element records. Most
power spectra of trace element profiles in pectinids show peaks associated with multi-day tidal
periodicities, the most dominant being a 7-day period, with weaker expression of cyclicity
associated with the fortnightly (14d) cycle or lunar month (28d). The latter is partly suppressed by





the 0.2 span LOESS filter (equivalent to a 44-56 day period depending on the length of the record)
applied on the records to remove the seasonal trend from the records. However, these lower
frequency cycles are clearly visible in the wavelets (see **S9**).
A much more consistent expression of diurnal periodicity is found in the tridacnid trace element
profiles compared to those in the pectinids (**Fig. 4**). Especially Sr/Ca and Ba/Ca records through
nearly all tridacnid specimens exhibit strong (>90% confidence level) power in the daily period,
while Mn/Ca and Mg/Ca records exhibit much less periodicity. Sr/Ca records in the tridacnids also
contain a significant (>96%) semi-diurnal component, whose tidal origin seems clear in most
specimens by peaks in power in the longer (7d, 14d and 28d) tidal components.


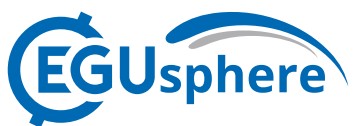

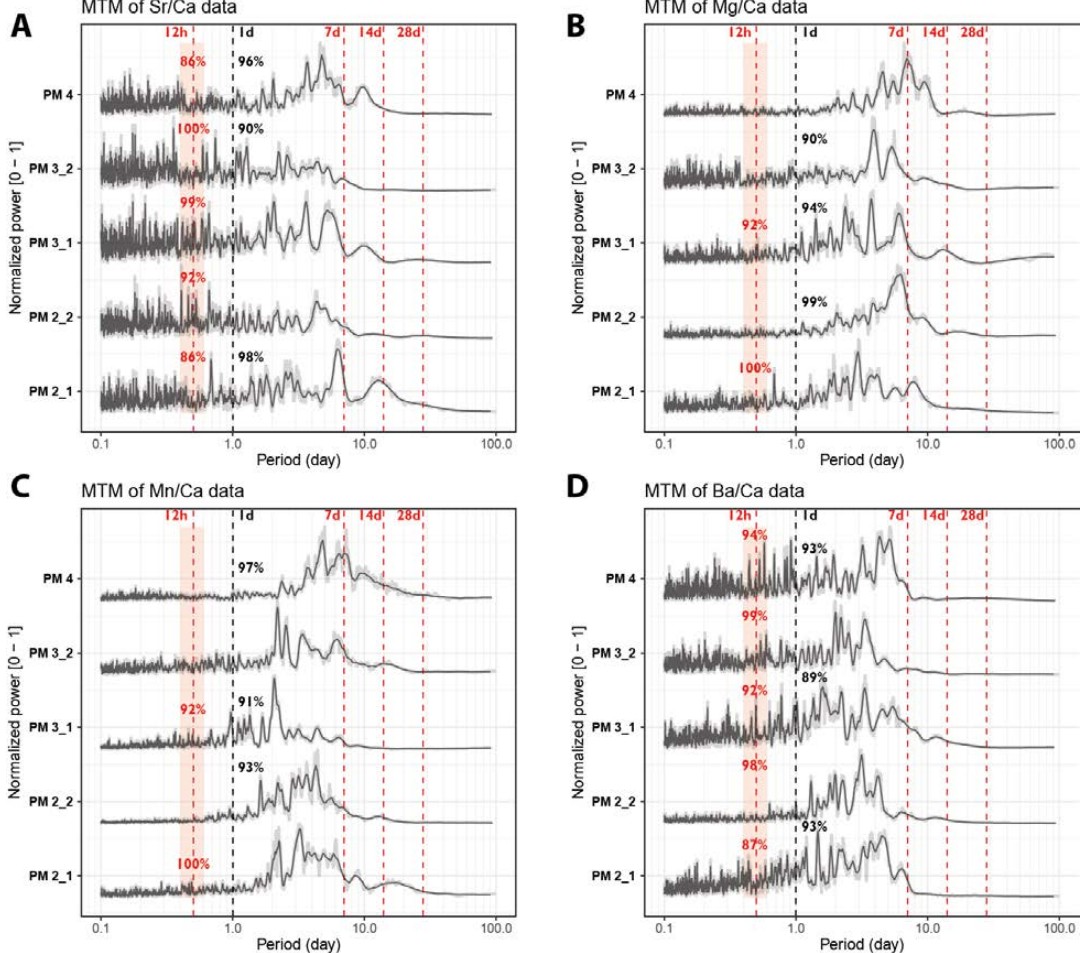


**Figure 3:** Multi-taper method spectrograms of Sr/Ca (**A**), Mg/Ca (**B**), Mn/Ca (**C**) and Ba/Ca (**D**)
records through the five pectinid cross sections after detrending (see **section 2.6**). All spectra are
normalized by dividing by the highest power peak and plotted on the same horizontal axis. Grey
shaded lines show raw data while solid black lines plot 21-point moving average smoothed
curves. Red vertical dashed lines highlight the expected periods of tidal variability while black
vertical dashed lines indicate 1-day periodicities. Significance levels of peaks on these periods
(see **section 2.6** and Meyers, 2012) are rounded to the nearest whole percentage point.





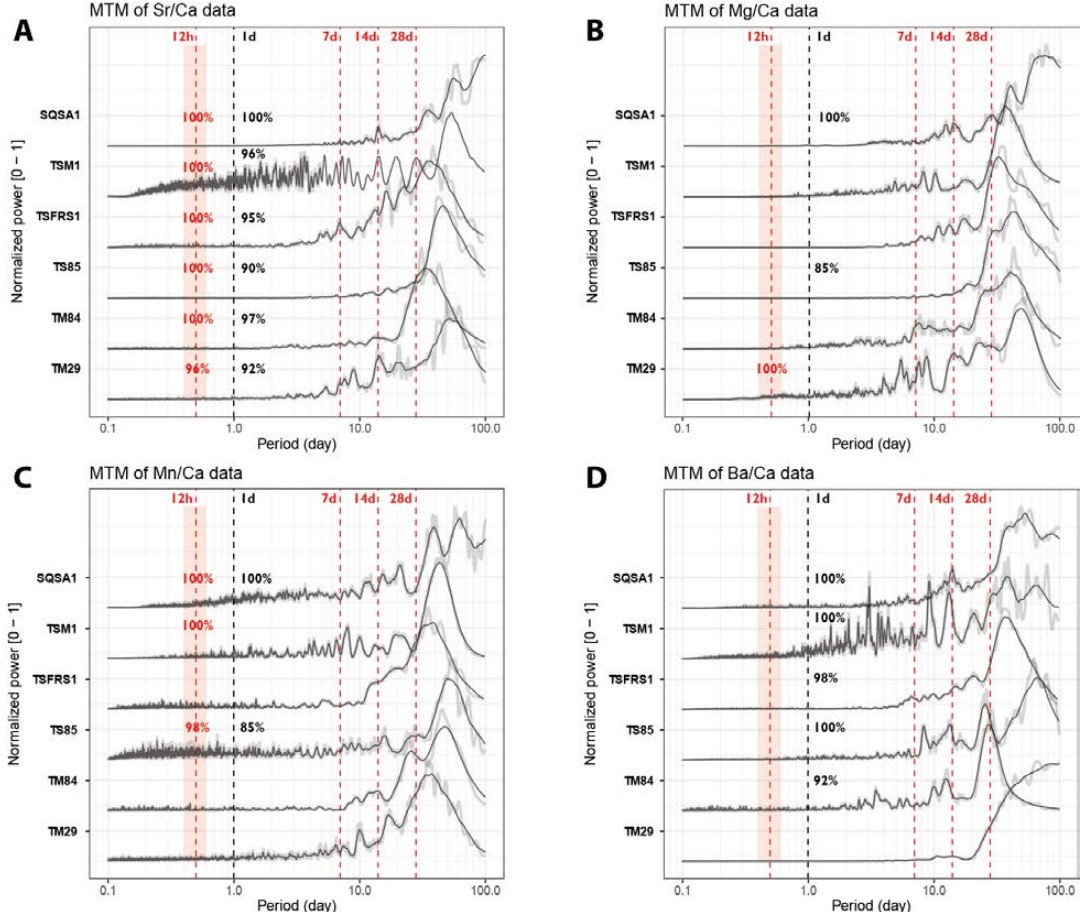

450

**Figure 4**: Multi-taper method spectrograms of Sr/Ca (**A**), Mg/Ca (**B**), Mn/Ca (**C**) and Ba/Ca (**D**) records through the six tridacnid cross sections after detrending (see **section 2.6**). All spectra are normalized by dividing by the highest power peak and plotted on the same horizontal axis. Grey shaded lines show raw data while solid black lines plot 21-point moving average smoothed curves. Red vertical dashed lines highlight the expected periods of tidal variability while black vertical dashed lines indicate 1-day periodicities. Significance levels of peaks on these periods (see **section 2.6** and Meyers, 2012) are rounded to the nearest whole percentage point.





3.4 Variance decomposition
Variability at the daily (24h) and tidal (12h) scale in all trace element records through all specimens
was extracted using bandpass filtering (**section 2.7;** see **S9** and **S10**). The median amplitude of
variability within these stacks was plotted against the median period of the variability per element
and per specimen to highlight dominant periodicities in the trace element data (**Figure 5**). As
noted in the spectral analysis results (**section 3.3**), trace element composition in tridacnid shells
is more strongly controlled by daily variability than in pectinid shells (**Fig. 5**; **S10**). The difference
is especially noticeable in Sr/Ca and Ba/Ca ratios, which show a clear divide between daily
periodicity in tridacnid shells and tidal periodicity in pectinids (see **Fig. 5**). The differences in
Mg/Ca and Mn/Ca ratios are less clear.






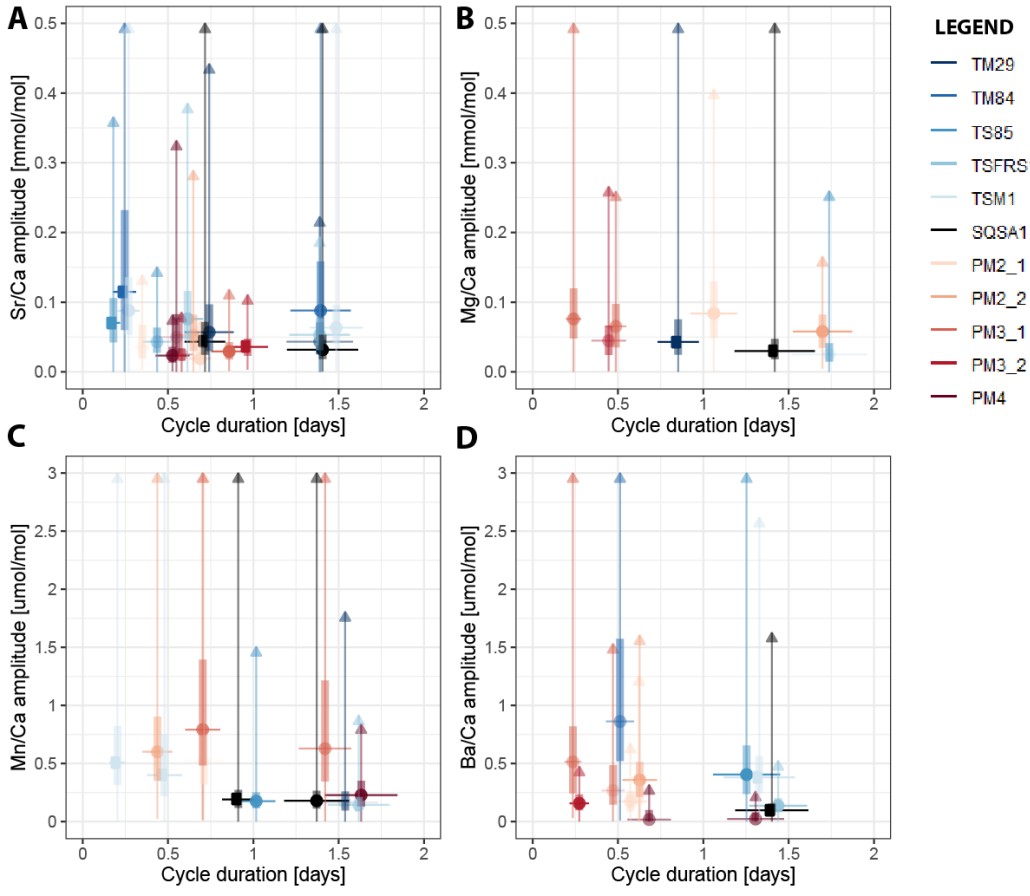

**Figure 5:** Cross plot showing the amplitude of variability of dominant spectral periods in Sr/Ca

(**A**), Mg/Ca (**B**), Mn/Ca (**C**) and Ba/Ca (**D**) against the period (duration) of the cycle. Round

symbols indicate the median amplitude of the cycle, while vertical bars and lines show interquartile

differences and ranges in the amplitude over the record. Horizontal bars indicate the width of the

bandpass filter used to extract periodic variability. Colors highlight different specimens (see

legend).





An example of the distribution of normalized variability within the trace element records after each
data processing step is shown in **S7**. From this example it is clear that a large fraction of the
variance in the records (73% in this record after trimming outliers) is explained by low-frequency
variability (**S7**). Of the remaining smoothed and detrended dataset, at most 20% of the variance
is explained by daily and tidal (semi-diurnal) periodicity (see **Figure 6** and **Table 3**). A full
decomposition of variance in all trace element records through all specimens is provided in **S7**.
**Figure 5** and **Figure 6A** shows that, overall, the variance explained by daily periodicity is higher
in tridacnids than in pectinids (Wilcoxon signed rank test; W = 44; p = 0.009). The difference
between species is smaller for tidal variability (**Fig. 6B**). There is no clear difference in relative
dominance of tidal variability between trace element records, but daily variability is more strongly
expressed in Ba/Ca and Mn/Ca records, especially in tridacnid shells. Finally, *T. squamosa*
specimen **TSM1**, which grew under a sunshade, does not exhibit significantly lower daily
periodicity compared to the other tridacnid specimens.




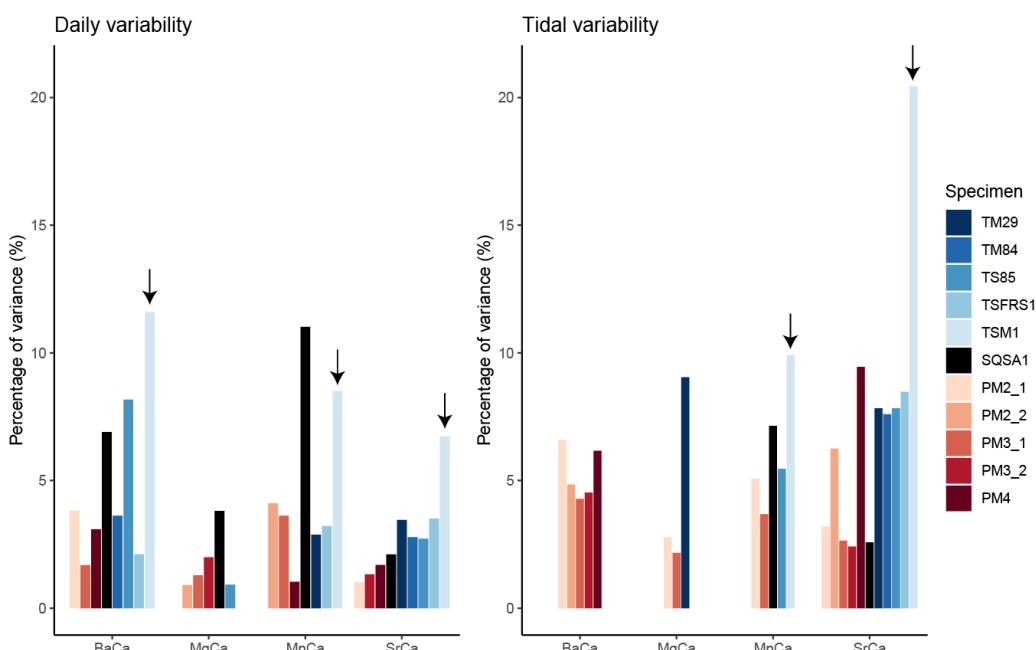


**Figure 6:** Summary of relative variance (in %) of significant daily (left) and tidal (right) variability

extracted from trace element records. Colors highlight different specimens (see legend). Note that

the *T. squamosa* specimen **TSM1** which grew under a sunshade is highlighted with a black arrow.




**Table 3**: Overview of the relative (in %) variance associated with daily and tidal variability in all
trace element records through all specimens. Empty cells represent records for which no
significant tidal or daily periodicity was found (see **Fig. 3-4**).

| | Daily variance [% relative to detrended record] | | | | Tidal variance [% relative to detrended record] | | | |
|---|---|---|---|---|---|---|---|---|
| | Ba/Ca | Mg/Ca | Mn/Ca | Sr/Ca | Ba/Ca | Mg/Ca | Mn/Ca | Sr/Ca |
| **PM2_1** | 3.8 % | | | 1.0 % | 6.6 % | 2.8 % | 5.1 % | 3.2 % |
| **PM2_2** | | 0.9 % | 4.1 % | | 4.9 % | | | 6.3 % |
| **PM3_1** | 1.7 % | 1.3 % | 3.6 % | | 4.3 % | 2.2 % | 3.7 % | 2.7 % |
| **PM3_2** | | 2.0 % | | 1.3 % | 4.5 % | | | 2.4 % |
| **PM4** | 3.1 % | | 1.0 % | 1.7 % | 6.2 % | | | 9.5 % |
| **TM29** | | | 2.9 % | 3.5 % | | 9.0 % | | 7.8 % |
| **TM84** | 3.6 % | | | 2.8 % | | | | 7.6 % |
| **TS85** | 8.1 % | 0.9 % | | 2.7 % | | | 5.5 % | 7.8 % |
| **TSFRS1** | 2.1 % | | 3.2 % | 3.5 % | | | | 8.5 % |
| **TSM1** | 12 % | | 8.5 % | 6.7 % | | | 10 % | 20 % |
| **SQSA1** | 6.9 % | 3.8 % | 11 % | 2.1 % | | | 7.1 % | 2.6 % |






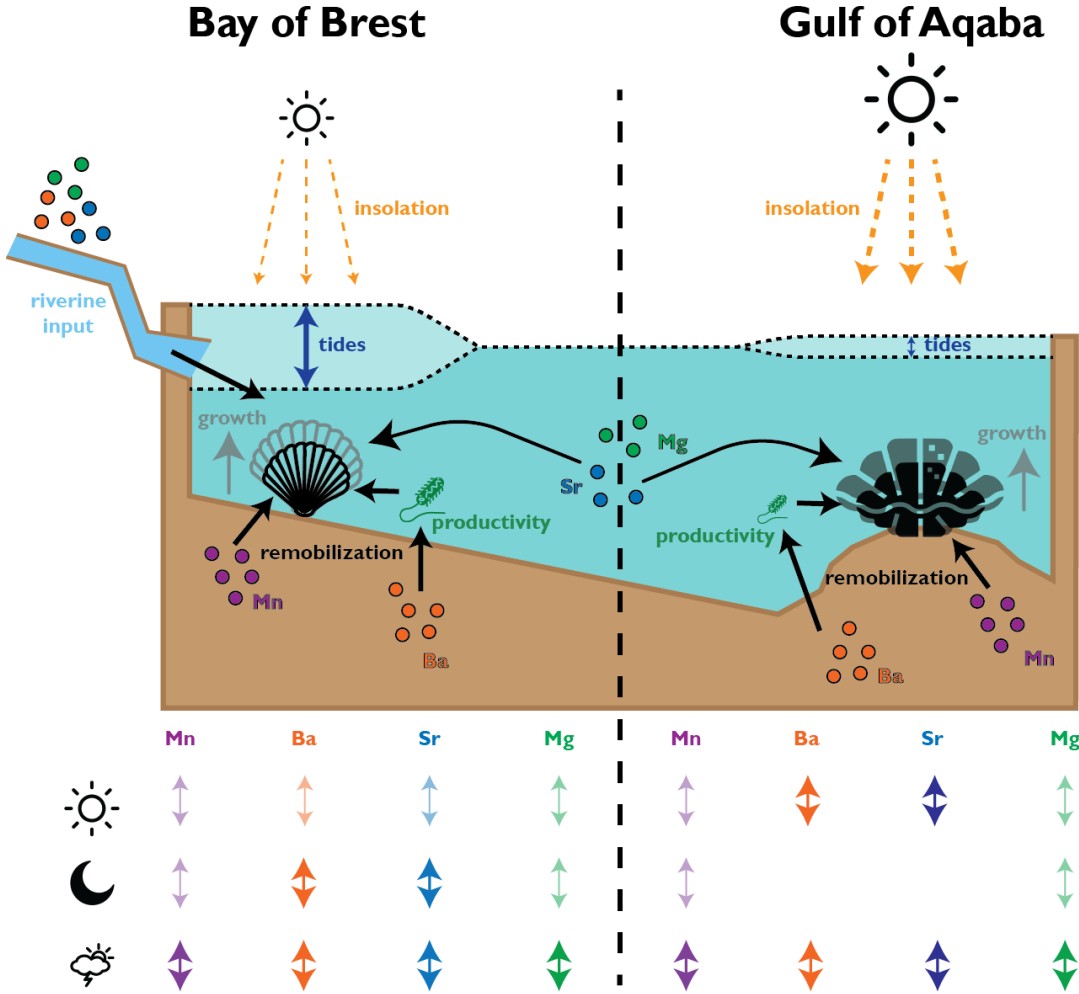

**Figure 7**: Schematic overview of environmental parameters interpreted to affect shell growth and composition of pectinids in the Bay of Brest and tridacnids in the Gulf of Aqaba. The table at the bottom provides a schematic qualitative overview of the amount of variance in the trace element records of the taxa is explained by daily (sun symbol), tidal (moon symbol) or aperiodic (cloud symbol) variability in the environment.





**4. Discussion**
4.1 Trace element variability in *P. maximus*
4.1.1 Comparison with previous studies
Trace element concentrations in *P. Maximus* analyzed in this study are in close agreement with
concentrations found in wild (live collected) *pectinid* shells in the literature (Lorrain et al., 2005;
Barats et al., 2008; Poitevin et al., 2020; Fröhlich et al., 2022). In these studies, Sr/Ca shows a
strong link with calcification rate (as measured by the width of daily shell increments; Lorrain et
al., 2005), although previous studies did not assess variability on the (sub-)daily scale. The long-
term trends in our Sr/Ca records seem to confirm this correlation, with higher values being
recorded in the middle of the growing season (day 50-150; **Fig. 2**) when growth rates are highest
(see **S5**). There is some discussion on the dependence of Mg/Ca ratios in pectinid shells to
temperature and/or salinity (Lorrain et al., 2005; Poitevin et al., 2020). This study's individuals that
grew during the same year in the same environment do not show a synchronous Mg/Ca pattern
(**Fig. 2**), arguing against a simple temperature dependence for Mg/Ca in *P. maximus.* In addition,
the lack of strict coherence between profiles of Mg/Ca (and other elements) in parallel transects
through *P. maximus* shells (e.g., **PM2_1** and **PM2_2**; **Fig. 2**) hints at compositional heterogeneity
within the shells. Low correlations between profiles through the same shell at the daily scale are
also partly driven by small misalignments of the timing of shell formation between the transects
at the sub-millimeter scale and variations in the height of trace element peaks, especially in Mn/Ca
and Ba/Ca, which are higher further towards the outside of the shell (**S2**). There is evidence
suggesting that Mg content varies in mollusk shells in function of the amount of organic matter in
the biomineral (Dauphin et al., 2003; Richard, 2009; Tanaka et al., 2019). Contrarily, Mn is taken
up in thermodynamic equilibrium in the mineral fraction of bivalve shells (Onuma et al., 1979;
Soldati et al., 2016), and Mn/Ca ratios in *P. maximus* have been shown to faithfully record
fluctuations of dissolved Mn in the coastal environment driven by riverine input and redox





conditions (Barats et al., 2008). Similarly, there is strong evidence that Ba/Ca ratios in *P. maximus*
(and other mollusks) record changes in Ba available in the environment linked to primary
productivity (e.g., Gillikin et al., 2008; Thébault et al., 2009; Fröhlich et al., 2022). This relationship
would explain the skewed (skewness > 1; **S8**) character of the Ba/Ca records and the correlation
between Ba/Ca and Mn/Ca in our *P. maximus* specimens, as the reducing conditions following
peaks in primary productivity favor the remobilization of Mn into the water column causing short-
lived simultaneous increases in Ba/Ca and Mn/Ca in the shells (Dehairs et al., 1989; Barats et al.,

540    2008; 2009).

4.1.2  Short-term changes in shell composition in tridacnids
In the context of the high-resolution trace element variability central to this study, it seems
plausible that short-term changes in the environment of the Bay of Brest were drivers of Mn/Ca
and Ba/Ca variability in *P. maximus* shells, while Mg/Ca and Sr/Ca composition is mostly driven
by changes in calcification rate. This would suggest that the significant tidal (12h) component in
Ba/Ca and Mn/Ca records through *P. maximus* (**Fig. 3**) is driven directly by redox changes over
the strong tidal cycle in the Bay of Brest (see Polsenaere et al., 2021) and resuspension of Ba
and Mn due to tidal currents (Hily et al., 1992), while tidal rhythms in Mg/Ca and Sr/Ca may be a
consequence of the scallop's calcification response to changes in its environment (e.g.,
temperature, salinity and light availability) through the large (up to 7m range) tidal cycle (**Fig. 7**).
The latter corroborates with previous studies in other calcitic mollusk shells which demonstrated
that Mg incorporation on short timescales is driven by the metabolic response to subtle changes
in the environment (Lazareth et al., 2007). Finally, care must be taken to interpret trace element
variability in *P. maximus* shells, since large intra-shell gradients in Mg/Ca, Sr/Ca and Mn/Ca have
previously been observed in this species, making trace element composition highly dependent on
the location of measurements relative to the outer shell surface or positioning relative to striae on
the shell surface (Freitas et al., 2009). Even though the LA-ICP-MS line scans in this study





targeted exclusively the oOSL of *P. maximus* specimens, variability in elemental ratios resulting
from small changes in the distance of the line scan from the outer edge of the shell cannot be fully
excluded (Richard, 2009).

4.2 Trace element variability in tridacnids
4.2.1 Comparison with previous studies
Results for Sr/Ca, Mg/Ca and Ba/Ca in this study's tridacnid specimens broadly corroborate trace
element results in other tridacnid studies (e.g., Elliot et al., 2009; Sano et al., 2012; Yan et al.,
2013; Warter et al., 2018). While data on Mn/Ca in the OSL of tridacnids is scarce, the Mn/Ca
ratios in tridacnids in this study (mean Mn/Ca = 7.8 ± 4.7 µmol/mol) are similar to LAICPMS Mn/Ca
data available in the literature (Warter et al., 2015, 4-10 µmol/mol), but significantly lower than
Mn/Ca values measured using total digestion Atomic Absorption Spectrometry (Madkour, 2005,
~30 µmol/mol). The main difference between the techniques is that LA-ICP-MS (both in this study
as in Warter et al., 2015) sampled shell layers where growth lines were visible and did not include
pre-treatment, while the total digestion study (Madkour, 2005) removed organic matter by roasting
the shells at 200°C prior to bulk shell analysis. The difference in results may therefore hint at
differences between shell layers within tridacnids, or differences in Mn concentration between the
organic and mineral fractions in the shells. Bivalve typically contain between 1% and 5% organic
matter (Marin and Luquet, 2004), with tridacnid shells being notable for their low organic matter
content (<1%; Agbaje et al., 2017; Taylor and Layman, 1972). Given that most Mn in bivalve
shells is typically associate with the mineral fraction of the shell (Soldati et al., 2016), it seems
unlikely that such a large fraction of Mn could originate from the organic matrix. Therefore, we
consider a difference in Mn concentration between shell layers in tridacnids more likely. The lack
of consistent trace element offsets between the tridacnid species under study here (*T. maxima*,




*T. squamosa* and *T. squamosina*) confirms the chemical similarities of shells tridacnid species
found in previous studies (e.g., *T. gigas*; Elliot et al., 2009; Yan et al., 2013; *T. crocea*; Warter et
al., 2018; *T. derasa*; Sano et al., 2012).
4.2.2 Short-term variability in Sr/Ca
Sr/Ca in tridacnids is thought to be strongly controlled by light intensity through a circadian rhythm
linked to the day-night cycle (Sano et al., 2012; Warter et al., 2018). This would explain the strong
daily periodicity in Sr/Ca records through all tridacnids in this study. This daily periodicity may be
caused by the ctenidium in tridacnids working on a daily rhythm to keep the acid-base balance in
the hemolymph of the clams to offset the $CO_2$ depletion by photosymbionts (which is paced to the
day-night cycle of light availability). In the process, $Ca^{2+}$-channels and $Na^+/H^+$-exchangers work
to keep the charge balance in the internal fluid and provide nutrients and ions for shell
mineralization, letting in compatible trace elements such as $Sr^{2+}$ (Ip and Chew, 2021). This
mechanism could explain the indirect link between trace element uptake in the shell in tridacnids
and the day-night cycle without a direct causal relationship between trace element concentration
and light availability (as demonstrated by the strong daily cycle in trace elements in the shaded
**TSM1** specimen). It is worth noting that experiments on freshwater bivalves (e.g., *Corbicula*
*fluminea*; Zhao et al., 2017) revealed that a closure of the $Ca^{2+}$ channels did not influence Sr
concentrations in the shell, arguing against a kinetic effect on Sr partitioning into the shell.
4.2.3 Tidal vs. diurnal variability
Our spectral analysis does not allow us to distinguish between the expression of the solar day
(24h) and lunar day (~24.8h) because the width of the bandpass filters used to extract periodicities
encompass both frequencies. While we cannot exclude the possibility that some of the daily (24h
frequency band) periodicity in tridacnid records is an expression of the lunar cycle, it seems
unlikely for most records except Sr/Ca, because the expression of the other lunar cycles (most



notably the ~12h cycle) is much weaker in tridacnids compared to the pectinids. Nevertheless, it
remains possible that the diurnal cycle in Sr/Ca in tridacnids, previously interpreted as a response
to the day-night cycle, is in fact caused by a circadian rhythm paced to the lunar day. Additionally,
vertical mixing, a major driver of sea surface temperature changes in the northern Gulf of Aqaba
is shown to be driven by a combination of surface wind intensity (which has strong daily variability)
and the presence of tidal currents (Carlson et al., 2014). It is therefore possible that changes in
local surface water temperature partly control the observed (semi-)diurnal variability.
4.2.4 Seasonal variability
On longer (seasonal) timescales, Sr/Ca in tridacnids has been suggested as a temperature proxy
similar to the well-known Sr/Ca-Sea Surface Temperature relationship in tropical corals (Lough,
2010; Yan et al., 2013). However, significantly lower Sr/Ca ratios in tridacnids compared to coral
aragonite (1.5 – 2.0 mmol/mol vs. 7.5 – 9.5 mmol/mol in corals; Elliot et al., 2009; **Fig. 2**) suggest
that tridacnids exert a large degree of biological control on the Sr concentration in their shells,
either possibly through the light-sensitive photosymbiosis-calcification relationship outlined above
or otherwise through active Sr removal from the biomineralization site by Sr-binding organic
molecules. Similarly, Mg/Ca ratios in tridacnids were previously thought to primarily record water
temperature (e.g., Batenburg et al., 2011) but detailed investigation shows here large differences
in Mg concentration within tridacnid shells. and a strong anticorrelation of Mg with sulfur
compounds associated with the organic matrix in the shell (see **section 4.1**; Dauphin et al., 2003),
has been put forward as evidence for a strong control of calcification and microstructure on Mg
composition in tridacnid shells (Yoshimura et al., 2014). However, evidence from studies on
foraminifera calcification demonstrate that the sulfur in biocarbonates is not organically bound
and that the covariation with Mg might instead be caused by lattice distortion due to incorporation
of Mg favoring simultaneous S incorporation (van Dijk et al., 2017).
4.2.5 Ba/Ca and Mn/Ca in tridacnids





As in the pectinids, Ba/Ca ratios in tridacnids likely reflect changes in Ba in the environment, which
can be caused by river input, upwelling of comparatively nutrient-rich waters or blooms of Ba-rich
phytoplankton (Vander Putten et al., 2000; Elliot et al., 2009). Given that Mn is mostly associated
with the mineral fraction of bivalve shells and seems to fractionate into the shell close to
equilibrium with seawater (Onuma et al., 1979; Soldati et al., 2016), Mn/Ca ratios in tridacnids
likely reflect the availability of dissolved Mn in the water column, as in other mollusk taxa (e.g.,
Barats et al., 2008; see **section 4.1**). This assumption is supported by the correlation between
Mn/Ca and Ba/Ca measured in this study (**Fig. 2**), suggesting that both records are influenced on
seasonal timescales by variability in nutrient availability and redox conditions (*sensu* Dehairs et
al., 1989). Part of this correlation between Mn/Ca and Ba/Ca is driven by synchronous increases
in both elements near the start and end of the profiles through tridacnid shells (**Fig. 2**). These
changes may reflect a decrease of control on shell composition during periods of stress, or
alternatively reflect periods of slower growth which cause more primitive microstructures
(characterized by higher concentrations of trace elements) to be formed (Warter et al., 2018).
4.2.6 Environmental changes in the Gulf of Aqaba
Given that the Gulf of Aqaba is oligotrophic, seasonally stratified, and lacks significant riverine
input (Nassar et al., 2014; Manasrah et al., 2018), the variability in nutrient concentrations and
redox conditions driving Mn/Ca and Ba/Ca variability in tridacnids are likely driven by convective
overturning. The tidal amplitude is much smaller than in the Bay of Brest (<1 m; Manasrah et al.,
2018) and is unlikely to drive significant short-term fluctuations in sea water chemistry. This may
therefore explain the lack of tidal (12h) periodicity in Ba/Ca and Mn/Ca in tridacnids (**Fig. 5**) .
Nevertheless, tidal rhythms have been observed in the behavior and growth of deep-sea bivalves
living far below the direct influence of tides on the environment, proving that such patterns can be
recorded by the animals through their circadian rhythm (Schöne and Giere, 2005; Nedoncelle et
al., 2013; Mat et al., 2020). In this case, the daily cycle seems to have been more important for





Ba/Ca in tridacnids, plausibly by driving diurnal changes in primary productivity in the Gulf of
Aqaba. Alternatively, the daily periodicity found in tridacnid shell chemistry may in fact be a
response to the lunar day (~24.8h) cycle, which is imprinted in the shell's chemical composition
through periodic exposure of the clams to extreme heat or air (subaerial exposure) in their shallow
water environment during exceptionally low tides. The stress induced from this exposure could
have affected calcification and incorporation of trace elements (see above).
Interestingly, Sr/Ca ratios in tridacnids do exhibit tidal periodicity (**Table 2**), perhaps driven by a
circadian rhythm in calcification linked to the tidal cycle, or by subtle changes in water temperature
driven by tidal currents (Carlson et al., 2014). This 12h periodic behavior is not observed in
previous studies of Sr/Ca ratios in tridacnids (Sano et al., 2012; Warter et al., 2018). A recent
valvometric study on tridacnids found a 12h period in activity, which supports the hypothesis that
a circadian rhythm paced to the tidal cycle could influence shell calcification (Killam et al., 2022).
Significant daily fluctuations in solar radiation (up to 1500 W m$^{-2}$; Manasrah et al., 2018) likely
exerted a dominant control on the calcification of tridacnids, explaining the strong diurnal
periodicity in Sr/Ca and Ba/Ca records in this study (see **Fig. 7** and **Fig. 9**). As in the (non-
symbiotic) pectinid data, it seems likely that the majority of Mn/Ca and Ba/Ca variability in
tridacnids directly reflects changes in the chemistry of the sea water and its constituents (e.g.,
particulate organic matter) while Mg/Ca and Sr/Ca variations are driven by changes in calcification
and microstructure. The latter may be indirectly influenced by light intensity through
photosynthesis by the symbionts, or by circadian rhythms paces to the diurnal or tidal cycle.

4.3 Role of photosymbiosis on high-frequency chemical variability
4.3.1 Effect of symbiosis on calcification





While the amplitude of diurnal variability in trace element concentrations does not vary much
between the symbiotic tridacnids and the non-symbiotic pectinids (**Fig. 5**), the amount of variance
in the trace element records explained by daily cyclicity is up to twice as high in tridacnids (**Fig.**
**6**; **Table 3**). This suggests that the 24h cycle has a much larger relative influence on trace element
composition (especially Sr/Ca and Ba/Ca) in tridacnids than in pectinids. This seems to point
towards a role of the photosymbionts in calcification by tridacnids, such as through symbiont-
mediated diurnal variation in the pH of the extrapallial fluid (Ip et al., 2006), as well as active
transport of $HCO_3^-$ for calcification (Chew et al., 2019) and as a C supply to the symbionts from
the host (Boo et al., 2021). Given the differences in absolute ratios between these two groups of
bivalves, comparing variance yields a more robust assessment of the relative importance of tidal
or diurnal variability on shell composition than looking at the absolute size (amplitude) of the
chemical cycle. While the difference in variance is clear, the importance of diurnal cyclicity on the
photosymbiotic tridacnids is not as big as one might expect. Rarely more than 10% of the variance
is explained by day-night variability (**Table 3**). This seems to contradict the large daily Sr/Ca
amplitudes found in Warter et al. (2018) and the trace element fluctuations found in de Winter et
al. (2020), which rival the seasonal cycle in these trace element ratios in terms of amplitude.
However, the percentages in **Table 3** relate to the amount of variation in the complete records
through these individuals and therefore also contain areas of the shell where daily cyclicity is less
pronounced, while values in previous studies often reflect maximum amplitudes recorded in parts
of the shell with exceptionally clear daily increments.
4.3.2 Effect of differences in the environment
It seems plausible that part of the difference in diurnal variability between pectinids and tridacnids
is explained by a difference in the environment between the Gulf of Aqaba and the Bay of Brest,
rather than the presence of photosymbionts. The diurnal insolation cycle in the Gulf of Aqaba is
larger than in the Bay of Brest (1500 vs 546 W*m$^{-2}$ maximum summer irradiance; Roberts et al.,





2018; Manasrah et al., 2019). If calcification in pectinids would be equally sensitive to sunlight,
this difference may explain much of the difference between the species. In this scenario, part of
the strong tidal component in the pectinid trace element data may be explained by the influence
of differences in water depth on the penetration of sunlight through the murky waters of the Bay
of Brest (Roberts et al., 2018). In fact, tidal movement can cause strong non-linear amplification
or reduction of the solar irradiance at the sea floor of the Bay of Brest by factors exceeding 10,
especially outside the summer months, which in turn has a significant effect on primary
productivity in the water column (Roberts et al., 2018). This tidal effect is likely to be much weaker
in the Gulf of Aqaba, given its comparatively low tidal amplitude, clear oligotrophic waters, and
much stronger and less seasonal day-night cycle (Manasrah et al., 2019). Indeed, even in non-
photosymbiotic bivalves, light and food availability are demonstrated to be major drivers of the
animal's behavior (e.g., Ballesta-Artero et al., 2017). The combination of the daily and tidal cycles
on solar irradiance at depth and photosynthesis in the Bay of Brest may therefore pose an
alternative pathway for strong tidal cyclicity in the trace element composition of pectinids in this
study and account for part of the twofold increase in daily variability in tridacnids compared to the
pectinids (**Fig. 6-7**; **Table 3**).
4.3.3 Effect of direct insolation
Specimen **TSM1** poses an interesting case study for investigating the link between sunlight and
calcification in tridacnids, since it grew under a sunshade and therefore experienced a dampened
diurnal variability in insolation compared to other giant clams in the area. The fact that this
specimen exhibits similar or even higher diurnal variability in shell chemistry (**Fig. 6**) argues
against a direct influence of the rate of photosynthesis itself on calcification. Instead, it seems that
daily chemical variability is mostly an expression of circadian rhythm in tridacnids, which is
strongly (evolutionarily) coupled to the day-night cycle to optimize the symbiosis with primary
producers in its mantle, possibly through respiration rhythms carried out by the ctenidium (see



**section 4.2**; Ip and Chew, 2021). Symbionts have been shown to directly aid in calcification in
terms of proton pumping (Armstrong et al., 2018), influencing internal acid-base chemistry (Ip et
al., 2006), and valvometric studies show the clams bask in sunlight in daylight hours and close
partially at night when symbiosis is likely reduced (Schwartzmann et al., 2011). This conclusion
is further supported by the lack of a clear difference in diurnal cyclicity between trace element
records in *T. maxima*, *T. squamosa* and *T. squamosina* (**Fig. 6**; **Table 3**), even though the degree
of reliance on photosymbiosis is demonstrated to be highly variable between these species
(Killam et al., 2020). Therefore, it seems unlikely that sub-daily resolved trace element records in
tridacnids can be used as quantitative recorders of paleo-insolation, as was originally suggested
by Sano et al. (2012). While the degree of symbiotic activity may not be clearly recorded in the
daily amplitude of trace element oscillations, the greater consistency of daily periodic signal in the
studied giant clams could relate to the direct biological control exerted by the symbionts on the
hosts' rhythms of calcification. Light exposure in giant clams promotes expression of genes coding
for proteins involved $Ca^{2+}$, $H^+$ and $HCO_3^-$ transport in the mantles of giant clams (Ip et al., 2017;
Chew et al., 2019), with the expression proposed to be at least partially mediated by photosensing
on the part of the symbionts themselves (Ip et al., 2017). Differences between the daily
consistency (spectral power) of photosymbiotic and non-photosymbiotic trace element profiles
might still allow paleontologists to use the presence of strong daily periodicity as a proxy for
photosymbiosis in the fossil record (as suggested in de Winter et al. 2020). However, the small
differences found between pectinids and tridacnids in this study and the comparatively large
influence of environmental variability show that such records should be interpreted with caution.
Future studies could measure photosynthetic activity of the symbionts in tridacnids and attempt
to relate this to the trace element composition of the shell in an attempt to isolate the direct effect
of photosymbiosis on shell composition.





4.4 Aperiodic drivers of shell chemistry
4.4.1 Circadian and behavioral changes
Even after controlling for instrumental noise, most (~90%) of the variance observed in our trace
element records is not directly related to the diurnal or tidal cycle. This suggests that aperiodic
events at the scale of hours to days play an important role in the calcification of pectinids and
tridacnids. Given the large difference in ecological niche (e.g., photosymbiotic versus non-
symbiotic) between these taxa, and the difference between the environment in which they grew,
this observation suggests that calcification of bivalves at the (sub-)daily scale may generally be
dominated by aperiodic variability in calcification or in the environment. Part of this unaccounted
variability may be caused by variability in the animal's behavior, as documented by observations
of siphon and valve gape activity in cultured or monitored specimens of a variety of bivalve taxa
(Rodland et al., 2006; Ballesta-Artero et al., 2017). While these experiments revealed quasi-
periodic (3-7 minute and 60–90 minute periods) behavior unassociated with the tidal or daily cycle,
records of activity of the individuals also reveal less regular patterns on the scale of 2-24h which
may contribute to the aperiodic variance in trace element records (Rodland et al., 2006). Another
example of aperiodic behavior potentially influencing shell chemistry is rapid valve adduction or
"coughing" observed in both pectinids and tridacnids, which serves as a mechanism for expelling
respiratory $CO_2$ and faeces from the pallial cavity or to evade predation attempts (Robson et al.,
2012; Soo and Todd, 2014). This behavior could resuspend sediment and produce pulses of Mn
and Ba at the sediment-water interface which are recorded as short-term, aperiodic variability in
these elements in the shell. The temporal sampling resolution (>1h) of trace element records after
smoothing out measurement noise does not allow us to resolve periodic variability at the sub-
hourly scale cited in these previous studies, meaning that aperiodic variability in behavior and
aliasing of these ultradian patterns likely contribute to the aperiodic variability in our trace element
records. On longer (sub-)seasonal timescales, activity in bivalves is shown to be highly dependent





on food and light availability (Ballesta-Artero et al., 2017), suggesting that aperiodic, short-term
changes in these environmental factors could be a main driver of shell growth and composition
and explain a large part of the variance in the trace element records which is not explained by
ultradian changes in the animal's behavior.
4.4.2 Short-term environmental changes and paleoweather
Outside of regular fluctuations caused by tidal, daily and seasonal cycles, changes in light and
food availability at the hourly to daily scale are probably linked to circulation and weather
phenomena. Previous studies show that enhanced vertical mixing during weather events such as
storms, algal bloom events after wind-driven upwelling and pseudoperiodic dust deposition can
temporarily increase the concentration of dissolved metals in surface waters, resuspend organic
matter and temporarily increase primary productivity. (Lin et al., 2003; Al-Najjar et al., 2007; Iluz
et al., 2009; Al-Taani et al., 2015; Komagoe et al., 2018). This will in turn lead to a shallowing of
the redoxcline through increased organic matter load at the sediment-water interface, which can
be recorded in the composition of giant clam shells (Yan et al., 2020). Interestingly, data in Yan
et al. (2020) suggest that recording an extreme weather event in *Tridacna* requires wind speeds
exceeding 20 km/h, a threshold which is almost never reached in the comparatively quiet Gulf of
Aqaba (Manasrah et al., 2019), while such events are common in the stormier Bay of Brest (Hily
et al., 1992; Chauvaud et al., 1998). This highlights another difference between the environments
of pectinid and tridacnid specimens investigated in this study which could contribute to the
variable expression of periodicity in the trace element composition of their shells. A plausible
scenario therefore emerges in which aperiodic weather events cause short-term variability in both
the chemistry and physical properties of the water column. These changes are subsequently
recorded in bivalve shells, either directly because the weather events resuspend, remobilize or
deliver trace elements like Mn and Ba (e.g., Dehairs et al., 1989; Gillikin et al., 2008; Mahé et al.,
2010), or indirectly because environmental stress associated with the event affects behavior and





shell calcification, resulting in a change in the incorporation of alkali-group cations (e.g., Mg and
Sr) into the shell biomineral (Carré et al., 2006; Takesue et al., 2008; **Fig. 7**). Our results therefore
highlight the potential of high-resolution trace element records in bivalve shells to record short-
term circulation changes and weather events, while prescribing caution in interpreting such
records until the effect of true environmental changes on the sub-daily scale can be separated
from aperiodic ultradian or behavioral patterns.



## 5. Conclusions

Our high-resolution trace element records reveal that short-term variability on the tidal or daily scale is recorded in the Mg, Sr, Mn, and Ba composition of shells of fast-growing mollusk species. The application of spectral analysis and variance decomposition on these trace element records is a useful tool to assess the influence of periodicity in the shallow marine environment on calcification in mollusk shells. Our statistical analysis reveals that tidal and daily variability each account for less than 10% of trace element variance in pectinids and tridacnids. In photosymbiotic giant clam shells, the amount of variance in Sr and Ba paced to the daily cycle is two times higher than in the non-photosymbiotic pectinids, suggesting that photosymbiosis in giant clams exerts some control on trace element composition in their shells. However, since only ~10% of the trace element variability in tridacnids is explained by diurnal variability, the recognition of photosymbiosis in the fossil record from diurnal variability in fossil shell composition will be complicated. In addition, differences between the mid-latitude environment of the pectinids and the tropical environment of the tridacnids likely account for part of the difference in trace element composition between the taxa. We propose that Ba and Mn composition in pectinids and tridacnids reflect short-term variability in primary productivity and sea water chemistry which control the mobility of these elements. Concentrations of Mg and Sr are likely controlled by short-term changes in growth and metabolic rate of the mollusks, which may be indirectly controlled by changes in their environment through circadian rhythms or behavior, explaining the pacing of trace element composition to the tidal and diurnal cycle. Most of the variance in trace element records in both taxa are not related to periodic behavior at the 12h or 24h scale, likely recording aperiodic events in the environment related to weather-scale phenomena or circadian patterns. We thus conclude that mollusk shell carbonate is a promising archive for recording weather-scale variability in shallow marine environments across latitudes, potentially recording weather-scale



phenomena in deep time, as long as these environmental effects can be separated by the
influence of the animal's behavior.

**Code availability**
Scripts used for data processing and to create figures in this manuscript were uploaded to an
open-access repository on GitHub (https://github.com/nielsjdewinter/TE_circadian) and linked
through Zenodo (https://zenodo.org/record/6603175).

**Data availability**
Supplementary data and figures referenced in this contribution were uploaded to the online open-
access repository Zenodo (https://doi.org/10.5281/zenodo.6602894).

**Author contribution**
NJW designed the experiment after discussion with BRS, DK and LF. LF, DK, BRS and JT
collected the samples. LF, DK and NJW together prepared samples for analyses and constructed
shell chronologies using growth line counting. WB, LN, GJR and NJW carried out the LA-ICP-MS
analyses and data processing. NJW designed and carried out the statistical analyses and wrote
the R scripts guided by feedback from LN, BK, LN, WB and GJR. NJW wrote the first draft of the
manuscript. All authors contributed to the writing process towards the final version of the
manuscript.

**Competing interests**



The authors declare that they have no conflict of interest.

**Acknowledgements**

The authors would like to thank Leonard Bik for his help with sample preparation and Maarten Zeilmans for his help with high-resolution imaging of the samples at Utrecht University. This study is part of the UNBIAS project, jointly funded by a Flemish Research Foundation (FWO; 12ZB220N) post-doctoral fellowship (NJW) and a MSCA Individual Fellowship (H2020-MSCA-IF-2018; 843011 – UNBIAS; awarded to NJW). GJR and LKD acknowledge funding from the Netherlands Earth System Science Center (NESSC; grant no. 024.002.001) from the Dutch Ministry for Education, Culture and Science (gravitation grant no. NWO 024.002.001). BRS acknowledges funding from the Deutsche Forschungsgemeinschaft (DFG; SCHO/793/21). JT acknowledges funding from the French National Research Agency (ANR; ANR-18-CE92-0036-01) awarded within the framework of the French-German collaborative project HIPPO (HIgh-resolution Primary Production multiprOxy archives).

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
