# Peer review of "Ultradian rhythms in shell composition of photosymbiotic and non-photosymbiotic"

_EGUsphere, 2022_

## Referee Comment (RC2)

**Review of**

**De Winter et al. **Ultradian rhythms in shell composition of photosymbiotic and non-photosymbiotic mollusks**

*Submitted for publication in Biogeosciences*

In general, this is a very interesting manuscript on a topic that has generated quite a few contributions in the past few years, not only in molluscs but also – even earlier – in foraminifera tests. The issue at stake is what controls the observed (sub)daily chemical variability, here specifically in mollusc shells. This is what De Winter et al. set out to resolve by comparing spatially-resolved chemical signals in molluscs with and without photosymbionts, namely tropical giant clams (*Tridacna, T*) vs scallops (*Pecten, P*) that also live in strongly contrasting tidal regimes. This is an interesting, actually quite nifty approach that has the potential of significantly contributing to the issue at stake.

The manuscript is quite long, overall well written besides key issues identified below, and contains extensive, almost too extensive (several GB of data!) SI. It presents LA-ICP-MS data of 8 specimens (3x P, 5x T) plus corresponding age models, followed by spectral analysis - and overall aims to identify the nature and cause of high-resolution, i.e. sub-daily elemental variability.

However, I'm afraid to say that without significant additional documentation, their current dataset – especially valid for slower growing giant clams T – is **not** capable of revealing sub-daily compositional signals. Thus, it is hard to see how their careful, elaborate evaluation via spectral analysis etc can be upheld. I'll detail this below.

So, a more detailed evaluation of the manuscript and its implications has to wait until this documentation has been provided or these issues have been clarified. Hence, this review does not necessarily cover all aspects, as it strikes me necessary to iron our input data first before making further, potentially far-reaching interpretations.

The main issues to be addressed include:

**1) Spatial vs. temporal vs. sampling resolution of LA-ICP-MS data:** The earlier papers on daily-resolved geochemical cycles in Tridacna by Sano and co-workers used a NanoSIMS at 2 µm spatial resolution, which subsequent LA-ICP-MS work at 3-4 µm by e.g. Warter et al tried to achieve as well. Previous work by De Winter et al (2020) used 10 µm spots (circular, rectangular). Thus using a rectangular slit of 100 x 20 µm, 20 µm in growth direction, hardly counts as ultra-high spatial resolution (L 107), and it crucially is insufficient to achieve hourly resolution (L107) in many of the samples investigated, chiefly the Tridacnas. According to their Tab. 1, Pecten grow ~250 µm/day and thus 20 µm indeed nominally represent ~2 h. However, their subtropical counterparts, Tridacna, only grow between 22-40 µm/day (Tab. 2). It's the comparison between the two groups that represents the overall aim of the authors, so the slowest growing ones do matter a lot. They address this issue on p25 (L415-419) and state that they achieve *"…resolution of the LAICPMS data (0.4 µm)…"*. However, every LA spot averages over 20 µm or possibly more as there is also the lateral dimension of 100 µm to be considered - due to the laser-sampling at 20 µm (L417). The (nominal) 0.4 µm resolution (L415) comes from the interaction between sweep time (0.1 s) and laser scan speed (4 µm/s; all in Tab.11), which **at best is the sampling**

**resolution of an individual data point, but NOT what can be resolved temporally in clams that grow 20-40 µm/day and are being analyzed with a 20 µm laser spot.**

So, I'm sceptical that they can achieve 50-fold better (=20/0.4) temporal resolution, allowing them to claim (L412): *[…] average temporal resolution of the LA-ICP-MS line scans was 0.04h, 0.24h, 0.44h and 0.27h for […]*. But this is crucial for their spectral analysis where they require data at sub-daily time resolution.

I thus invite the authors to show comparative plots, at very high spatial resolution, namely ~400 µm for Tridacnas and ~2000 µm for Pecten - that reveal compositional cycles at sub-daily resolution (similar to the work by Sano or Warter etc.) based on their current dataset. The data shown in Fig. 2 do not provide this level of detail at all. I did not find a data table that would have allowed me to re-plot the data myself.

Once this is achieved, the data can be reassessed with respect to the implications of their spectral analyis. To be honest, I doubt that their existing dataset will reveal such sub-daily cycles due to the insufficient combination of laser-spot size and laser scan speed, but maybe I'm missing something and thus this should be added. If this is not possible, then the samples may need to be re-analyzed with much smaller laser spot sizes and slower scan speed.

**2) LA-ICP-MS data:** While there is overall good documentation, it is necessary to get data documenting accuracy from MACS-3 and BAS752 and JCp1. Please provide some general details for BAS752 as this is less well-known standard material. I doubt that the sweep time (run cycle time) is 100 ms, given that the sum of all 6 m/z is exactly 100 ms, and some time is spent between the masses. Why was B not analysed, given that B/Ca can show very well-resolved daily cyclicity in giant clams?

**3) Definition of terminology:** What exactly is meant by 'semi-diurnal'? Does it mean half daily (12 h=tidal?) or *approximately* daily? Semi may mean half or approx. Please define.

**4) Shell growth – a few issues:** How useful is it to utilize maximum shell height ($L_{inf}$) from the literature since the authors did growth band counting and interpolation in between? How does one unequivocally identify growth breaks visible on the outer margin? Doesn't the statement (L257) " […] distinction between diurnal (24h) and tidal (~12h) pacing of growth increments […] imply some form of circular reasoning?

In Tab. 2, e.g. TS85, how does a diurnal width of 40.3 µm correspond to an annual width of 20.2 mm? On my reckoning, it is 14.7 mm.

**5) Results overall:** Keep the results description to a minimum overall, refer to figures and tables upfront, and move sections such as the comparison between P & T (~L327-L347) to the discussion. These are calcitic vs aragonitic shells, so differences are to be expected simply based on $K_d$'s. What is a 'typical' seasonal pattern (L350) – this is again mixing results with interpretation, which has to be avoided. L346 – don't mix ratio with concentration presentation and use quantitative rather than qualitative comparative statements (L376, 377).

L319: Instead of a fixed value of 0.05 mmol/mol, such differences should be given as %deviations.

**6) Results of spectral analysis:** Fig. 3 (Pecten) – even for these fast growing clams, the respective peaks at daily and half-daily (12 h) timing are not very clearly resolved. Instead, what is the meaning of the peaks between 1 d and 7 d? And in Fig. 4 I find no convincing peaks for the Tridacnids that indicate <~7 day periodicities, so I do NOT understand where the assertion in L436 is derived from. Hence my worry about temporal resolution of the LA-ICP-MS input data raised upfront!

**7) Further issues:**

a) Fig. 1: While it is a good figure in general, two issues should be changed. A-K is mixed between the two groups of organisms, and more importantly, I'd prefer to see much larger images that showcase the LA-ICP-MS profiles. B, C, K are too small and don't give sufficient detail.

b) Fig. 5: The content of this figure could be better assessed if we saw truly daily-resolved data, see above. Same for Fig. 6. If there is crucial information in some SOM-Figs, then move them into the main text please.

c) In L574-576 there is a certain amount of contradiction to previous statements about Mn incorporation.

d) L588 The authors did not resolve daily periodicity in Sr/Ca in tridacnids in my view, so they can't make statements like this.

**8) Referencing:** The references are in part incomplete with journal titles missing and others in Arabic font. Please proof read before submission. Killam et al 2022 missing.

**9) Minor issues:** This list is not comprehensive.

L120: 7.2 m – space between, here and elsewhere

L158 parallel

L327 'contain' is wrong wording, better 'are characterized'

L389 (Tab. 1): increment width – specify daily increment width;    and L399 (Tab. 2): what is semi-diurnal?

L502 Fig. 7 – good idea as a summary

L541 this appears misplaced here

Taken together this is an important study on a timely subject. The ideas conveyed in the abstract are broadly fine but can in detail not be assessed due to issues with the initial data raised above. I hope that this can be re-addressed. In its current form, the manuscript is not suitable for publication.

---

## Author Comment (AC1)

**EGUsphere-2022-576 - Reply to comments reviewer 1**

Dear reader,

We would like to thank Reviewer 1 for the positive review of our manuscript, and for the helpful comments that will allow us to improve it. Below, we reply to the review comments point-by-point and explain how we plan to make changes in the manuscript following the reviewer's suggestions.

de Winter et al. present a comprehensive dataset on trace elements in scallops and giant clams. The data are robust and they do a good job presenting them. My comments are minor and mostly deal with adding to the discussion and referencing previous works. I recommend to publish with minor changes.

We are glad to read that the reviewer thinks our contribution merits publication after minor revisions, and will do our best to revise our manuscript in reply to the issues raised below.

Abstract

Reads too positive – should be toned down.

We appreciate that we might have stated these claims too optimistically, and rephrased them as follows:

E.g.,:

"now enable the use of mollusk shells for paleoenvironmental reconstructions at a daily to sub-daily resolution"

Rephrased to: "now allows *in situ* determination of the composition of mollusk shell volumes precipitated at daily to sub-daily time intervals"

"We find significant expression of these periodicities"

Rephrased to: "We find weak but statistically significant expression of these periods, and conclude…"

L518-519 – be more explicit and less colloquial. "There is some discussion"? In the published literature? Maybe say previous studies… Then "This study's" – are you referring to the citations in the previous sentence or are you referring to the work you present here?

Also, "the same year in the same environment" as what?

L521 "arguing against a simple temperature dependence for Mg/Ca" – maybe say agreeing with previous studies

Following these suggestions, we rephrased the section in L518-521 as follows:

"Previous studies demonstrate that Mg/Ca ratios in pectinid shells are at most partially related to temperature and/or salinity (Lorrain et al., 2005; Poitevin et al., 2020). The fact the studied *P. maximus* specimens, which all grew during the same year in the same environment, do not show a synchronous Mg/Ca pattern (Fig. 2) agrees with previous work and argues against a simple temperature dependence for Mg/Ca in *P. maximus*."

L523 – Lorrain et al. 2005 also found this – state that here

Agreed, we rephrased to: "…hints at compositional heterogeneity within the shells, in agreement with findings by Lorrain et al. (2005)."

L536 Gillikin et al. 2006 (doi:10.1016/j.gca.2005.09.015) discuss separating the background Ba/Ca from the peaks. If these shells all grew in the same salinity they should all have similar background values. Do you see this? I think this should be commented on here.

Our LA-ICP-MS results do not show consistent background Ba/Ca values in all specimens grown in the same environment (Fig. 2), which would argue against the statement by Gillikin et al. (2005). We wish to comment on this by adding the following sentence in line 535, between "(Fröhlich et al., 2022)." and "This relationship":

"Interestingly, our results (**Fig. 2**) show that background Ba/Ca values are not equal in the shells of *P. maximus* and tridacnid specimens grown in the same environment. This contradicts the assessment by Gillikin et al. (2005) that background Ba/Ca concentrations are a function of environmental conditions and can be consistently subtracted from Ba/Ca records to separate peak from background values."

L541 – the subheader here is wrong, this section is about scallops

Correct, this should read: "Short-term changes in shell compositions in *P. maximus*" and will be rephrased accordingly.

L599 – should discuss Carre et al 2006 (already cited) and Gillikin et al. 2005 (doi:10.1029/2004GC000874) here.

We propose to refer here to the conceptual model for membrane permeability through $Ca^{2+}$-channels put forward by Carré et al. (2005), and argue against the hypothesis that Sr uptake is controlled by discrimination during mineralization from the extrapallial fluid into the shell (as proposed in Gillikin et al., 2005). This model is in better agreement with the most recent studies into tridacnid shell mineralization (cited in this paragraph). In the revised version, we conveyed this point by adding the following sentences after line 594:

"…such as $Sr^{2+}$ (Ip and Chew, 2021). This mechanism follows the biomineralization model by Carré et al. (2006) and is supported by the high affinity of $Sr^{2+}$ with Ca-channels (Hagiwara and Byerly, 1981) and the high ionic fluxes supported by this pathway, allowing enough membrane permeability to support the fast shell formation in tridacnids (Coimbra et al., 1988; Sather and McCleskey, 2003). Following this line of reasoning, the preconcentration of $Sr^{2+}$ in the extrapallial fluid through Ca-channels should have a larger effect on shell Sr/Ca ratios than the discrimination against $Sr^{2+}$ (or other trace elements) through shell organic matrix during mineralization of the shell from this fluid (as proposed in Gillikin et al., 2005). This model could explain the indirect link …"

Hagiwara, S. and Byerly, L.: Calcium Channel, Annual Review of Neuroscience, 4, 69–125, https://doi.org/10.1146/annurev.ne.04.030181.000441, 1981.

Coimbra, J., Machado, J., Fernandes, P. L., Ferreira, H. G., and Ferreira, K. G.: Electrophysiology of the Mantle of Anodonta Cygnea, Journal of Experimental Biology, 140, 65–88, https://doi.org/10.1242/jeb.140.1.65, 1988.

Sather, W. A. and McCleskey, E. W.: Permeation and selectivity in calcium channels, Annual review of physiology, 65, 133–159, 2003.

Gillikin, D. P., Lorrain, A., Navez, J., Taylor, J. W., André, L., Keppens, E., Baeyens, W., and Dehairs, F.: Strong biological controls on Sr/Ca ratios in aragonitic marine bivalve shells, Geochemistry, Geophysics, Geosystems, 6, https://doi.org/10.1029/2004GC000874, 2005.

Section 4.4.2 – I don't think this section clearly shows how your data contribute to this idea. It's mostly a discussion of previous studies. A sentence or two blending your results into this would bolster this discussion.

Agreed, we propose to rephrase the following segment to better explain how the effect of the processes we describe from the literature can be observed in our data:

We rephrased

"This highlights another difference between the environments of pectinid and tridacnid specimens investigated in this study which could contribute to the variable expression of periodicity in the trace element composition of their shells." (L796-798)

into

"This difference is also reflected in the periodicity of shell composition, with the tridacnids having overall higher percentages of their variance explained by daily and tidal variability than pectinids (**Fig. 6**), showing that aperiodic (potentially weather-controlled) variability in shell composition has a stronger influence on the pectinids which grew in the stormier Bay of Brest."

Later in the paragraph (L804-805), we already refer to our results and the proposed pathway by which variability in trace element composition we measured can be explained.

Many citations are missing journal names (e.g., L895, L903, L919, L924, L927, L929, L936 and many others…)

We will go through the reference list and add all missing information during our revision.

---

## Author Comment (AC2)

**EGUsphere-2022-576 - Reply to comments reviewer 1**

Dear editor,

We appreciate the time and effort spent by Reviewer 2 to comment on our manuscript. We are glad to read that the reviewer likes the design of our study overall and thinks the topic is worthwhile. Their main concern relates to the effective sampling resolution of our LAICPMS method and our ability to capture daily variability in the shells. Below, we provide a rebuttal to this major point below before listing our point-by-point replies to the other questions raised by the reviewer. We hope the changes we suggest will make our manuscript acceptable for revision and subsequent publication in Biogesciences.

In general, this is a very interesting manuscript on a topic that has generated quite a few contributions in the past few years, not only in molluscs but also – even earlier – in foraminifera tests. The issue at stake is what controls the observed (sub)daily chemical variability, here specifically in mollusc shells. This is what De Winter et al. set out to resolve by comparing spatially-resolved chemical signals in molluscs with and without photosymbionts, namely tropical giant clams (*Tridacna, T*) vs scallops (*Pecten, P*) that also live in strongly contrasting tidal regimes. This is an interesting, actually quite nifty approach that has the potential of significantly contributing to the issue at stake.

We are glad to read that the reviewer appreciates our study design and will include some references to earlier studies describing high-resolution chemical variability in foraminifera tests (e.g. Eggins et al., 2003; Anand and Elderfield, 2005).

Anand, P. and Elderfield, H.: Variability of Mg/Ca and Sr/Ca between and within the planktonic foraminifers Globigerina bulloides and Globorotalia truncatulinoides, Geochemistry, Geophysics, Geosystems, 6, https://doi.org/10.1029/2004GC000811, 2005.

Eggins, S., De Deckker, P., and Marshall, J.: Mg/Ca variation in planktonic foraminifera tests: implications for reconstructing palaeo-seawater temperature and habitat migration, Earth and Planetary Science Letters, 212, 291–306, https://doi.org/10.1016/S0012-821X(03)00283-8, 2003.

The manuscript is quite long, overall well written besides key issues identified below, and contains extensive, almost too extensive (several GB of data!) SI. It presents LA-ICP-MS data of 8 specimens (3x P, 5x T) plus corresponding age models, followed by spectral analysis - and overall aims to identify the nature and cause of high-resolution, i.e. sub-daily elemental variability.

We appreciate the feedback on the length of our manuscript and its supplements and will try to shorten it (especially the Results section) in the revision. However, we believe that the components in the supplement are necessary to ensure the reproducibility of our study, which is why we deposited them in an online repository where there is space for large files.

However, I'm afraid to say that without significant additional documentation, their current dataset – especially valid for slower growing giant clams T – is **not** capable of revealing sub-daily compositional signals. Thus, it is hard to see how their careful, elaborate evaluation via spectral analysis etc can be upheld. I'll detail this below.

So, a more detailed evaluation of the manuscript and its implications has to wait until this documentation has been provided or these issues have been clarified. Hence, this review does not necessarily cover all aspects, as it strikes me necessary to iron our input data first before making further, potentially far-reaching interpretations.

The main issues to be addressed include:

**1) Spatial vs. temporal vs. sampling resolution of LA-ICP-MS data:** The earlier papers on daily-resolved geochemical cycles in Tridacna by Sano and co-workers used a NanoSIMS at 2 µm spatial resolution, which subsequent LA-ICP-MS work at 3-4 µm by e.g. Warter et al tried to achieve as well. Previous work by De Winter et al (2020) used 10 µm spots (circular, rectangular). Thus using a rectangular slit of 100 x 20 µm, 20 µm in growth direction, hardly counts as ultra-high spatial resolution (L 107), and it crucially is insufficient to achieve hourly resolution (L107) in many of the samples investigated, chiefly the Tridacnas. According to their Tab. 1, Pecten grow ~250 µm/day and thus 20 µm indeed nominally represent ~2 h. However, their subtropical counterparts, Tridacna, only grow between 22-40 µm/day (Tab. 2). It's the comparison between the two groups that represents the overall aim of the authors, so the slowest growing ones do matter a lot. They address this issue on p25 (L415-419) and state that they achieve *"…resolution of the LAICPMS data (0.4 µm)…"*. However, every LA spot averages over 20 µm or possibly more as there is also the lateral dimension of 100 µm to be considered - due to the laser-sampling at 20 µm (L417). The (nominal) 0.4 µm resolution (L415) comes from the interaction between sweep time (0.1 s) and laser scan speed (4 µm/s; all in Tab.11), which **at best is the sampling resolution of an individual data point, but NOT what can be resolved temporally in clams that grow 20-40 µm/day and are being analyzed with a 20 µm laser spot.**

So, I'm sceptical that they can achieve 50-fold better (=20/0.4) temporal resolution, allowing them to claim (L412): *[…] average temporal resolution of the LA-ICP-MS line scans was 0.04h, 0.24h, 0.44h and 0.27h for […]*. But this is crucial for their spectral analysis where they require data at sub-daily time resolution.

We acknowledge that our explanation of the temporal resolution achieved with our LAICPMS method could be clarified, and we aim to do so in the revised version of our manuscript. We confirm the reviewer's analysis that every sample in our LAICPMS analysis averages an area with a width of 20 micrometer in scanning direction. The reported sampling resolution of 0.4 micrometer indeed comes from the combination of sweep time and scan speed, and this may have been confusing in the original manuscript. While we do achieve a roughly 0.4 micrometer resolution in terms of the number of datapoints per unit distance, we acknowledge that the use of a 20 micrometer wide spot causes smoothing of the record akin to applying a moving average. We hope to demonstrate that we can still pick up daily rhythms in our datasets using this approach by means of a virtual example:

Below, we provide some examples of how this smoothing would affect a theoretical banding consisting of daily cycles in El/Ca (e.g. Sr/Ca). We simulated this banding with a baseline Sr/Ca value of 1.5 mmol/mol and a diurnal amplitude of 0.6 mmol/mol (comparable to the amplitude found in e.g. Sano et al., 2012). The Sr/Ca record was projected on the distance domain using a Von Bertalanffy growth model of an artificial shell, which is constructed in the same way as the growth models for the tridacnid shells in this study (specifically specimen TM84 and SQSA1). We then virtually sampled from this dataset, simulating the full LA-ICP-MS procedure as follows: We sweep over the virtual record using our laser spot width of 20 micron, our scan speed of 4 micrometer/second and our cycle time of 109 ms (which includes ICP-MS dead time; see reply to comment below) and an integration time of Sr per ICP-MS sweep of 10 ms (see **S11**). We plotted the resulting record to demonstrate the smoothing that occurs due to our sampling method. We repeated this experiment with parameters for the growth of specimen TM84 (representing one of the faster-growing tridacnid specimens in our study) and SQSA1 (the slowest-growing specimen in our study and show the result in the plot below.

[Figure]

This experiment shows that, even in the slowest-growing tridacnid specimen in our study (SQSA1), this combination of 20 micron spot size and scan speed can pick up the daily resolution in the record and recover two thirds of the amplitude of the Sr/Ca signal. A signal like this would be picked up easily by the spectral analysis methods in our study, albeit with a slight reduction of the power value in the power spectrum (which is proportional to the square root of the amplitude). For the faster-growing tridacnid species, the amount of smoothing of the artificial record is nearly negligible. We hope these plots clarify the difference between our scanning resolution and the spatial smoothing due to our spot size and will add them, together with the script we used to make them, to the appendix of our revised manuscript as well.

In addition, please note that the median increment widths listed in Table 2 sometimes refer to semi-diurnal increments, meaning that the actual daily growth rate is twice the value listed in the table. The values listed in Table 2 are median values over the entire record, which contains variability in growth rate. Analyzing the entire growth period of the shell for (sub-)daily growth patterns and chemical variability inherently causes smoothing of the chemical variability due to averaging of the signal in places in the shell with strong variability with places in the record where (sub-)daily variability is less prominent. This caveat of our methodology is acknowledged in the original manuscript near the end of section 4.3.1. The spot size smoothing issue explained above adds to this smoothing effect and this will be acknowledged in the revised Discussion section. Note that the consideration of (sub-)daily variability over the entire growth period distinguishes our sampling strategy from that of the previous studies cited by the reviewer (Sano et al., 2012 and Warter and Müller, 2017), where an excerpt of the record is used to demonstrate the

presence of this variability. We believe our analysis better captures the variability in expression of ultradian rhythms over the entire growth period.

Sano, Y., Kobayashi, S., Shirai, K., Takahata, N., Matsumoto, K., Watanabe, T., Sowa, K., and Iwai, K.: Past daily light cycle recorded in the strontium/calcium ratios of giant clam shells, Nat Commun, 3, 761, https://doi.org/10.1038/ncomms1763, 2012.

Warter, V. and Müller, W.: Daily growth and tidal rhythms in Miocene and modern giant clams revealed via ultra-high resolution LA-ICPMS analysis—A novel methodological approach towards improved sclerochemistry, Palaeogeography, Palaeoclimatology, Palaeoecology, 465, 362–375, 2017.

I thus invite the authors to show comparative plots, at very high spatial resolution, namely ~400 µm for Tridacnas and ~2000 µm for Pecten - that reveal compositional cycles at sub-daily resolution (similar to the work by Sano or Warter etc.) based on their current dataset. The data shown in Fig. 2 do not provide this level of detail at all. I did not find a data table that would have allowed me to re-plot the data myself.

In the original manuscript, we decided not to add zoomed-in plots of the daily variability in the records because we aimed to discuss the expression of (sub-)daily variability in the entire growth period of the specimens rather than highlight parts of the shell where this variability is strongest (see reply above and discussion in section 4.3.1 of the original manuscript). In reply to this comment, we will add example plots of these high-resolution trace element rhythms in tridacnids and pectinids to show that this variability is present in our records. We also add two examples of such plots, one from *P. maximus* specimen 2 and one from *T. maxima* specimen 29, below to show what the (semi-)diurnal variability looks like when plotted in detail

In Tridacna shell 29, which prior work from Killam et al. (2021) suggested formed twice daily growth lines, data which we incorporated into our growth model, many of the days show "doublets," possibly aligning with the twice daily rise and fall of the tide, but as seen in the data, slightly out of phase with the modeled daily cycle, and not perfectly symmetrical, as the tides on a particular day aren't exactly equal.

We kindly note that all LAICPMS data required to produce the plots in this manuscript is provided in the online Zenodo repository (see Data Availability statement in manuscript for the link). We realize that this data was somewhat "hidden" in the large supplement and created a separate "raw data" folder at the top level of the supplement's folder structure to make it easier to find for the reader.

Killam, D., Al-Najjar, T., and Clapham, M.: Giant clam growth in the Gulf of Aqaba is accelerated compared to fossil populations, Proceedings of the Royal Society B: Biological Sciences, 288, 20210991, https://doi.org/10.1098/rspb.2021.0991, 2021.

[Figure]

*Zoomed-in plot of (semi-)diurnal variability in P. maximus specimen 2. The lines represent 51-point Savitzky-Golay filters showing the hour to daily scale variability while filtering out the higher-order measurement noise.*

[Figure]

*Zoomed-in plot of (semi-)diurnal variability in T. maxima specimen 29. The lines represent 21-point Savitzky-Golay filters showing the hour to daily scale variability while filtering out the higher-order measurement noise.*

Once this is achieved, the data can be reassessed with respect to the implications of their spectral analyis. To be honest, I doubt that their existing dataset will reveal such sub-daily cycles due to the insufficient combination of laser-spot size and laser scan speed, but maybe I'm missing something and thus this should

be added. If this is not possible, then the samples may need to be re-analyzed with much smaller laser spot sizes and slower scan speed.

**2) LA-ICP-MS data:** While there is overall good documentation, it is necessary to get data documenting accuracy from MACS-3 and BAS752 and JCp1. Please provide some general details for BAS752 as this is less well-known standard material. I doubt that the sweep time (run cycle time) is 100 ms, given that the sum of all 6 m/z is exactly 100 ms, and some time is spent between the masses. Why was B not analysed, given that B/Ca can show very well-resolved daily cyclicity in giant clams?

We will add details on the accuracy of LAICPMS values on the check standards in the appendix (**S12**). In summary, the accuracy on the matrix-matched MACS-3 carbonate standard is better than 5% for all elements in all LA-ICP-MS sessions from which data is used in this study.

The reference material called "BAS ECRM 752" is also known as "BAS-CRM 393" and is a well-known limestone standard (see https://rrr.bam.de/RRR/Content/EN/Downloads/RM-Certificates/RM-cert-iron-steel/RM-cert-ceramic-materials/b752_1e.pdf?__blob=publicationFile). We will rename it to "BAS CRM-393" in the text to avoid confusion.

Our reported ICP-MS sweep time indeed does not include the time needed for the ICP-MS to move between masses. The total cycle time including down time on the ICP-MS is 109 ms, and we add this to the supplement describing the LA-ICP-MS settings (**S11**).

The element B was not measured to limit the number of elements and keep the total cycle time as short as possible (maximizing the spatial resolution of the measurements).

**3) Definition of terminology:** What exactly is meant by 'semi-diurnal'? Does it mean half daily (12 h=tidal?) or *approximately* daily? Semi may mean half or approx. Please define.

Fair point, we define semi-diurnal as half-daily (=12h) and will make this explicit in the revised version of the manuscript.

**4) Shell growth – a few issues:** How useful is it to utilize maximum shell height (Linf) from the literature since the authors did growth band counting and interpolation in between? How does one unequivocally identify growth breaks visible on the outer margin? Doesn't the statement (L257) " […] distinction between diurnal (24h) and tidal (~12h) pacing of growth increments […] imply some form of circular reasoning?

In Tab. 2, e.g. TS85, how does a diurnal width of 40.3 µm correspond to an annual width of 20.2 mm? On my reckoning, it is 14.7 mm.

These are fair points, and we acknowledge that these aspects could have been explained more clearly in the manuscript. Initially, we aimed at expressing the full length of the tridacnid shells in counted lamina (as was done for the pectinids). The tridacnid shells were stained with Mutvei solution as described in Killam et al. 2021, to reveal internal growth lines. However, the stain did not make clear all growth lines visible through the growth record, which is a function of the organic material content of the shell. This varies through the ontogeny of the animal with relation to season and other factors. The identification of daily lines is still a field of active development in *Tridacna*, with some shells staining to reveal clear daily lines throughout the record (Komagoe et al., 2018), while others require more labor-intensive methods to reveal any daily lines at all (Liu et al. 2022).

Therefore, we opted for a hybrid method in which we measured the width of increments in a sample of the shells where they were well developed and used these measurements in combination with the annual growth breaks (which are easily recognized on the shell) to create age models for all specimens. This is also the reason why we needed to use literature values for $L_{inf}$ to anchor our growth model. Note that we

used well established $L_{inf}$ values from the same species in the same area to avoid biasing our growth model estimates (see section 2.5).

We do not believe that this method implies circular reasoning, since we inferred the (semi-)diurnal timing of the increments by comparing them with the independently measured annual growth breaks and use values for $L_{inf}$ from separate studies. In the revised version, we will attempt to better clarify our description of this methodology to take away the reviewer's concerns.

The apparent mismatch between (semi-)diurnal increment widths and mean annual growth (Table 2) stems from the fact that the former is the mean of the measured increments and the latter is the annual growth rate based on growth break distance. Since growth rates vary throughout the lifetime of tridacnids (both inter- and intra-annually) and these measurements were not necessarily made in the same part of the shell, the annual growth is not a simple multiple of the increment width.

Killam, D., Al-Najjar, T., and Clapham, M.: Giant clam growth in the Gulf of Aqaba is accelerated compared to fossil populations, Proceedings of the Royal Society B: Biological Sciences, 288, 20210991, https://doi.org/10.1098/rspb.2021.0991, 2021.

Komagoe, T., Watanabe, T., Shirai, K., Yamazaki, A., and Uematu, M.: Geochemical and Microstructural Signals in Giant Clam Tridacna maxima Recorded Typhoon Events at Okinotori Island, Japan, Journal of Geophysical Research: Biogeosciences, 123, 1460–1474, https://doi.org/10.1029/2017JG004082, 2018.

Liu, C., Zhao, L., Zhao, N., Yang, W., Hao, J., Qu, X., Liu, S., Dodson, J., and Yan, H.: Novel methods of resolving daily growth patterns in giant clam (Tridacna spp.) shells, Ecological Indicators, 134, 108480, https://doi.org/10.1016/j.ecolind.2021.108480, 2022.

**5) Results overall:** Keep the results description to a minimum overall, refer to figures and tables upfront, and move sections such as the comparison between P & T (~L327-L347) to the discussion. These are calcitic vs aragonitic shells, so differences are to be expected simply based on Kd's. What is a 'typical' seasonal pattern (L350) – this is again mixing results with interpretation, which has to be avoided. L346 – don't mix ratio with concentration presentation and use quantitative rather than qualitative comparative statements (L376, 377).

These are valid comments, and they echo the concern the reviewer voiced above about the length of our manuscript. During our revision, we will try to shorten the results section and integrate the indicated segments into the Discussion. We also double-checked our reference to element concentrations and ratios throughout this section.

L319: Instead of a fixed value of 0.05 mmol/mol, such differences should be given as %deviations.

We will calculate the relative deviations and provide these in the revised manuscript.

**6) Results of spectral analysis:** Fig. 3 (Pecten) – even for these fast growing clams, the respective peaks at daily and half-daily (12 h) timing are not very clearly resolved. Instead, what is the meaning of the peaks between 1 d and 7 d? And in Fig. 4 I find no convincing peaks for the Tridacnids that indicate <~7 day periodicities, so I do NOT understand where the assertion in L436 is derived from. Hence my worry about temporal resolution of the LA-ICP-MS input data raised upfront!

We tried to plot the spectral analysis results of these long, high-resolution records in many ways before settling on the current figure. The problem with plotting these is that the scale of the "power" value (y-axis) differs by more than an order of magnitude over the period domain (i.e. longer cycles have much higher power than shorter ones). This characteristic is inherent to spectral analysis, but causes the periodicities associated with shorter cycles (higher frequencies) to show much lower spectral power than the longer periods. We tried plotting these powerspectra on a log-log scale, but this compromised

recognition of peaks for the longer periods as well as on the shorter end of the spectrum. Therefore, we decided to show the confidence levels (percentage values) associated with peaks in the period domains highlighted in the figures. We fully agree that this is not the most intuitive way to plot these results, and we would have preferred to highlight the important peaks differently.

One thing we can try is to break the horizontal axis and highlight specific intervals of periodicity associated with the cycles of interest. In principle, breaking axes like this is not good plotting practice, but we are willing to try it to see if it is a better way to highlight the peaks in the powerspectra. If the reviewer has any alternative suggestions on how these results could be plotted in a way that ameliorates the large differences in power over the period domain, we would be very happy to hear them.

**7) Further issues:**

a) Fig. 1: While it is a good figure in general, two issues should be changed. A-K is mixed between the two groups of organisms, and more importantly, I'd prefer to see much larger images that showcase the LA-ICP-MS profiles. B, C, K are too small and don't give sufficient detail.

We appreciate this feedback and will make panels B, C and K bigger in the revised version. We assume that with the comment "A-K is mixed between the two groups of organisms", the reviewer means that the figure caption would be clearer if the letters were ordered by species (e.g. A-D for pectinids and E-I for tridacnids). We will implement this in the revised manuscript.

b) Fig. 5: The content of this figure could be better assessed if we saw truly daily-resolved data, see above. Same for Fig. 6. If there is crucial information in some SOM-Figs, then move them into the main text please.

In reply to these comments, we add excerpts of the records through tridacnids and pectinids to the manuscript showing examples of the (semi-)diurnal variability in the trace element ratios.

c) In L574-576 there is a certain amount of contradiction to previous statements about Mn incorporation.

We did not find contradiction in this section, but rephrased part of the discussion here to clarify our line of reasoning.

d) L588 The authors did not resolve daily periodicity in Sr/Ca in tridacnids in my view, so they can't make statements like this.

We respectfully disagree and hope to have demonstrated in reply to the major comment above that our methodology does resolve daily periodicity even in the slowest-growing tridacnids.

**8) Referencing:** The references are in part incomplete with journal titles missing and others in Arabic font. Please proof read before submission. Killam et al 2022 missing.

We thank the reviewer for pointing this out and will update and complete the reference section before submitting our revised manuscript version.

**9) Minor issues:** This list is not comprehensive.

L120: 7.2 m – space between, here and elsewhere

We will add the space throughout the manuscript

L158 parallel

Rephrased

L327 'contain' is wrong wording, better 'are characterized'

Agreed, this will be rephrased

L389 (Tab. 1): increment width – specify daily increment width; and L399 (Tab. 2): what is semi-diurnal?

We added "daily" and define "semi-diurnal" on first mention in the revised manuscript (see reply to comment above)

L502 Fig. 7 – good idea as a summary

Thanks!

L541 this appears misplaced here

Correct, this should read "Short-term changes in shell composition in pectinids" and will be rephrased.

Taken together this is an important study on a timely subject. The ideas conveyed in the abstract are broadly fine but can in detail not be assessed due to issues with the initial data raised above. I hope that this can be re-addressed. In its current form, the manuscript is not suitable for publication.

---

## Author Response (AR2)

Dear Niels and co-authors

First, apologies for the long time to reach a decision - I had hoped to secure a re-evaluation by one of the original reviewers yet this proved not to be possible.

You have transparently explained in excellent detail how both review reports were addressed, and I think the clarity of the manuscript has improved considerably, in particular regarding the reservations raised by Reviewer #2 regarding the resolution of your data (smoothing issues). This was the main concern and this is now well described in the revised version so that no ambiguity remains and readers can make their own correct assessment on how to interpret this.

I have listed a few minor changes below which I consider technical.

Line numbers refer to the revised (track changes) version

Dear Prof. Steven Bouillon,

Thank you for moderating the review process and for putting in the effort to try to get a second opinion from the reviewer. We highly appreciate you taking the time to go through our manuscript yourself to give us feedback and are glad to read that you consider our revisions acceptable. Below, we will briefly outline how we addressed your remaining points in a point-by-point fashion:

-throughout the text: use mmol mol-1, µmol mol-1, etc. rather than mmol/mol and µmol/mol, etc. This should be quick to replace.

This has been replaced throughout the text.

-L76-77: the oxygen isotope values --> the oxygen stable isotope composition

Rephrased

-L257 and L 266: remove '*'

We removed the "*"

-L473 (Table 1): please provide a more complete Table caption - should be clear without consulting the text. Unless I'm misinterpreting, the different values separated by comma's refer to counts by different people, and values in bold imply correspondence between increments counted on outer surface and in cross sections ?

Correct, these explanations have been added to the table caption.

-L653: (Lorrain et al., 2005) --> Lorrain et al. (2005)

The reference has been updated.

-L670: 'contradicts the assessment': would rephrase to indicate that the finding referred to would then not be generally valid.

Agreed, we rephrased to: "This suggests that the findings by Gillikin et al. (2005) that background Ba/Ca concentrations are a function of environmental conditions and can be consistently subtracted from Ba/Ca records to separate peak from background values may not be generally valid."

-L738:(Carré et al., 2006) --> Carré et al. (2006)

The reference has been updated, this consistent mistake has crept in due to the use of a reference editor and we have tried to amend it throughout the text.

-L745:(Gillikin et al., 2005) --> Gillikin et al. (2005)

The reference has been updated.

-L774: idem

The reference has been updated.

-L776: (Batenburg et al., 2011) --> Batenburg et al. (2011)

The reference has been updated.

-L1253-1254 (Madkour, 2005): reference contains Arabic; I doubt that was meant to be the case. Egyptian Journal of Aquatic Research, 31,45-59

This was indeed not intentional, and the reference has been updated.

-new Figure S13: might be more accessible when chosing a dotted line and full line instead of two line colors. I think this is an important Figure to illustrate your measurement approach and its effect on to the temporal signal of element ratios- you could consider making it part of the manuscript rather than supplement, but leave the choice to you.

We have amended the figure using a full line and dotted line to improve its accessibility, but decided to keep it as a supplementary figure given that the manuscript already contains a lot of figures and to prevent it from growing yet longer than it is.

In addition to the changes listed above in reply to the editorial comments, we took this opportunity to carefully read through the entire manuscript one more time and made several minor textual changes where we saw fit. All these changes were tracked in the new manuscript version using the "track changes" function in Microsoft Word.

Kind regards,

Niels de Winter

On behalf of the authors